# Source-specific light absorption by carbonaceous components in the complex aerosol matrix from yearly filter-based measurements

Vaios Moschos[1], Martin Gysel-Beer[1], Robin L. Modini[1], Joel C. Corbin[2], Dario Massabò[3], Camilla Costa[3], Silvia G. Danelli[3], Athanasia Vlachou[1], Kaspar R. Daellenbach[1], Sönke Szidat[4], Paolo Prati[3], André S. H. Prévôt[1], Urs Baltensperger[1], and Imad El Haddad[1]

[1]Laboratory of Atmospheric Chemistry, Paul Scherrer Institute, Villigen-PSI, CH-5232, Switzerland
[2]Metrology Research Centre, National Research Council Canada, Ottawa, ON K1A 0R6, Canada
[3]Department of Physics/(Industrial) Chemistry & INFN, University of Genoa, Genova, I-16146, Italy
[4]Department of Chemistry and Biochemistry / Oeschger Centre for Climate Change Research, University of Bern, Bern, CH-3012, Switzerland

*Correspondence to*: Imad El Haddad (imad.el-haddad@psi.ch)

Understanding the sources of light-absorbing organic (brown) carbon (BrC) and its interaction with black carbon (BC) and other non-refractory particulate matter (NR-PM) fractions is important for reducing uncertainties in the aerosol direct radiative forcing. In this study, we combine multiple filter-based techniques to achieve long-term, spectrally-resolved, source- and species-specific atmospheric absorption closure. We determine the mass absorption efficiency (MAE) in dilute bulk solutions at 370 nm to be equal to 1.4 $m^2 \, g^{-1}$ for fresh biomass smoke, 0.7 $m^2 \, g^{-1}$ for winter-oxygenated OA, and 0.13 $m^2 \, g^{-1}$ for other less absorbing OA. We apply Mie calculations to estimate the contributions of these fractions to total aerosol absorption. While enhanced absorption in the near-UV has been traditionally attributed to primary biomass smoke, here we show that anthropogenic oxygenated OA may be equally important for BrC absorption during winter, especially at an urban background site. We demonstrate that insoluble tar-balls are negligible in residential biomass burning atmospheric samples of this study, and thus could attribute the totality of the NR-PM absorption at shorter wavelengths to methanol-extractable BrC. As for BC, we show that the mass absorption cross-section (MAC) of this fraction is independent of its source, while we observe evidence for a filter-based lensing effect associated with the presence of NR-PM components. We find that bare BC has a MAC of 6.3 $m^2 \, g^{-1}$ at 660 nm and an absorption Ångström exponent (AAE) of $0.93 \pm 0.16$, while in the presence of coatings its absorption is enhanced by a factor of ~1.4. Based on Mie-calculations of closure between observed and predicted total light absorption, we provide an indication for a suppression of the filter-based lensing effect by BrC. The total absorption reduction remains modest, ~10-20 % at 370 nm, and is restricted to shorter wavelengths where BrC absorption is significant. Overall, our results allow an assessment of the relative importance of the different aerosol fractions to the total absorption, for aerosols from a wide range of sources and atmospheric ages. When integrated with the solar spectrum at 300-900 nm, bare BC is found to contribute around two thirds of the solar radiation absorption by total carbonaceous aerosols, amplified by the filter-based lensing effect (with an interquartile range, IQR, of 8-27 %), while the IQR of the contributions by particulate BrC is 6-13 % (13-20 % at the rural site during winter). Future studies that will directly benefit from these results include: (a) Optical

modelling aiming at understanding the absorption profiles of a complex aerosol composed of BrC, BC and lensing-inducing

coatings; (b) Source apportionment aiming at understanding the sources of BC and BrC from the aerosol absorption profiles; (c) Global modelling aiming at quantifying the most important aerosol absorbers.

**Keywords:** brown/black carbon, multi-wavelength absorption, source apportionment, optical closure, lensing suppression

## 1 Introduction

Short-lived climate forcers emitted from anthropogenic activities, e.g., residential wood burning, are ubiquitous in the atmosphere (Pöschl, 2005; Fuzzi et al., 2015). The net cooling effect of particle emissions has been partially masking the warming effect of long-lived greenhouse gases (Ramanathan and Feng, 2009). Nevertheless, a fraction of carbonaceous aerosols may lead to atmospheric heating (Ramanathan and Carmichael, 2008), but their radiative forcing remains uncertain (IPCC, 2013).


Black carbon (BC) is a strong broadband absorber with a nearly wavelength-independent refractive index and negligible solubility in common solvents (Bond and Bergstrom, 2006; Moosmüller et al., 2009; Petzold et al., 2013; Samset et al., 2018; Corbin et al., 2019). Organic aerosol (OA) also comprises light-absorbing compounds, collectively known as brown carbon (BrC). There exist various forms of BrC, including conventional soluble molecules absorbing in the near-UV region, and

insoluble tar carbon with appreciable absorption extending to the infrared region (Corbin et al., 2019). These forms may exert a positive radiative forcing on the climate potentially rivaling that of BC (Feng et al., 2013; Wang et al., 2014), especially over biomass burning-influenced regions (Liu et al., 2014b) and at higher altitudes (Zhang et al., 2017).

Accounting for the optical absorption properties of BrC, its interaction with BC, and their mixing state with other particulate

species are important for reducing the large uncertainty in the aerosol direct radiative forcing (Laskin et al., 2015; Gustafsson and Ramanathan, 2016; Samset et al., 2018; Saleh, 2020). While it is typically assumed that primary biomass burning dominates the BrC absorption, the importance of other sources with different absorption properties including secondary OA, formed through the oxidation of precursors, may be overlooked. A growing body of evidence suggests that aerosol aging may be associated with both bleaching and/or formation of secondary BrC (Saleh et al., 2013; Zhong and Jang, 2014; Zhao et al.,

2014; Kumar et al., 2018; Dasari et al., 2019). However, the net effect of these processes on the atmospheric BrC absorption remains elusive. Further, it is not yet established whether BrC from common urban sources is extractable, or else insoluble, refractory BrC can also contribute to total absorption.

Various in-situ measurement techniques, e.g., the single-particle soot photometer and the photo-acoustic aerosol absorption

spectrometer, have made significant progress in reporting BC absorption enhancement due to the so-called lensing effects by

coating materials (Moosmüller et al., 2009; Cappa et al., 2012; Lack et al., 2012; Pokhrel et al., 2017; Zanatta et al., 2018). Meanwhile, filter-based techniques, e.g., particle soot absorption photometer (PSAP) and Aethalometer (Hansen et al., 1984; Drinovec et al., 2015), report multi-wavelength light attenuation through a particle-laden filter that has to be calibrated in order to approximate the atmospheric absorption (Moosmüller et al., 2009; Müller et al., 2011). Recently, the Aethalometer has

become the method of choice when simplicity, low price, widespread deployment and unattended robust operation over long time periods in the field are sought. An Aethalometer-based source apportionment (SA) model (Sandradewi et al., 2008), hereafter denoted "the Aethalometer SA model", has been proposed to separate the contributions of wood burning and traffic emissions to equivalent BC, eBC (Petzold et al., 2013). The Aethalometer SA model is based on differences in the spectral profiles between the two aerosol sources, with biomass burning emissions characterized by enhanced absorption at shorter

wavelengths, or high absorption Ångström exponents (AAE). Various upgraded versions of this approach were introduced thereafter (Favez et al., 2010; Herich et al., 2011; Martinsson et al., 2017; Zotter et al., 2017) and this approach is currently widely used for eBC source apportionment. However, it is unclear how this model responds to photochemical aging, SOA formation and lensing (Martinsson et al., 2015; Garg et al., 2016; Dasari et al., 2019).

Two main approaches exist to estimate the absorption by individual aerosol components in heterogeneous atmospheric particle ensembles (Moffet et al., 2010). The first approach is based on online measurements and consists of a direct estimation of BrC absorption at shorter wavelengths by subtracting the estimated total BC absorption (assuming an AAE for bulk BC) from the total aerosol absorption. This rather convenient approach might lead to biased estimates (see Sect. 4), if the decoupling from potentially variable optical properties of pure BC and its absorption enhancement due to coating acquisition is not ensured.

Alternatively, isolation of extractable BrC absorption by filter solvent extraction requires additional calculations for the conversion to atmospheric absorption, which are not straightforward (Liu et al., 2013; Shetty et al., 2019) because assumptions on BC/BrC particle size and mixing state are needed (see Sect. 2.1.4 and Sect. 2.3). Biomass burning BC, unlike fresh traffic BC (Cappa et al., 2012), is expected to be at least partially internally-mixed with co-emitted BrC (Schwarz et al., 2008; Lack et al., 2012; Liu et al., 2014a; Liu et al., 2017; Shamjad et al., 2017). With aging, BC from both sources may be further coated

with non-refractory particulate matter species (BrC, non-absorbing OA, inorganics). This internal mixing typically results in an absorption enhancement of up to 1.5 at longer wavelengths (Fierce et al., 2016; Yuan et al., 2020). At shorter wavelengths, the lensing effect is theoretically predicted to be dampened due to the absorption of BrC (Lack and Cappa, 2010; Saleh et al., 2015; Luo et al., 2018). However, experimental evidence for lensing suppression has yet to be provided.

In this study, we report the total carbonaceous aerosol absorption based on yearly filter-based measurements from two sites in Switzerland. We determine the absorption of total solvent-extractable BrC and apply statistical methods to assign this absorption to different OA classes following the approach of Moschos et al. (2018). In addition, we derive the absorption of bare BC and its wavelength dependence, and assess experimentally the influence of particulate BrC and its effect on filter-based lensing.

## 2 Materials and methods

A schematic of the methodology, including the available instruments and datasets, is shown in Fig. S1.

### 2.1 Measurements

#### 2.1.1 Measurements of aerosol species mass concentrations

Atmospheric aerosols of two size fractions (310 samples in total; 245 $PM_{10}$ (particulate matter with an aerodynamic diameter $d \leq 10$ μm) and 65 $PM_{2.5}$ ($d \leq 2.5$ μm) were collected in Magadino and Zurich (Switzerland) during 2013-2014 on quartz fiber filters for laboratory measurements (Text S1). Magadino is a rural background site affected in winter by intense wood burning activity for residential heating, whereas Zurich is an urban background site affected by regional transport of anthropogenic-dominated pollution in winter and spring. Both sites are affected by biogenic secondary emissions in summer (see Text S1; Fig. S2). The daily filter samples were analyzed for the mass concentrations of elemental and organic carbon (Sunset EC/OC), water-soluble organic carbon (WSOC; measured with a total carbon analyzer) and secondary inorganic ions (SIA = $NO_3^-$, $SO_4^{2-}$, $Cl^-$ and $NH_4^+$; major-SIA = $NO_3^-$ and $SO_4^{2-}$) by ion chromatography. Selected filters from Magadino were also measured for radiocarbon ($^{14}C$) in the EC fraction using the Swiss4s protocol (Zhang et al., 2012), to determine the fossil fraction of EC ($EC_{fossil}$). Factor contributions from different aerosol sources to water-soluble and total OA (Fig. S2) mass concentrations $M$ (in μg m$^{-3}$) were available for $PM_{2.5}/PM_{10}$ samples from Magadino (Vlachou et al., 2018) and $PM_{10}$ samples from Magadino and Zurich (Daellenbach et al., 2017), based on offline aerosol mass spectrometer (Daellenbach et al., 2016) analysis coupled with positive matrix factorization (offline AMS/PMF). Hereafter we refer to their PMF results as "Solution 1" and "Solution 2", respectively. The determination of $M$ was based on total OA (OC × OA/OC; OC from Sunset and OA/OC from offline AMS) and water-insoluble OA [WINSOA = (OC - WSOC) × OA/OC]. The OA source components (factors) resolved by offline AMS/PMF may be related to primary emissions from traffic (HOA), cooking (COA), biomass burning (BBOA), vehicular/tire wear (sulfur-containing, SCOA), and of biological aerosols (PBOA), as well as non-fossil secondary oxygenated aerosol in winter (WOOA) and in summer (SOOA) and from fossil precursors (fOOA). As explained below (Sect. 3.2), factors with low contribution to absorbance, including HOA, COA, SOOA and fOOA, were combined into one "Other OA" factor for each AMS/PMF solution.

#### 2.1.2 Measurements of particle size distributions and morphology

The particle size distribution was continuously measured in Zurich using a scanning mobility particle sizer (SMPS, 10 min. time resolution). Daily-averaged data were considered (Text S2) for optical calculations. A field-emission scanning electron microscope coupled to energy-dispersive x-ray spectroscopy (FE-SEM/EDS) was used (Text S2) to observe the general morphology and deposition characteristics of particles, and to provide information on their elemental composition.

### 2.1.3 Measurement of total aerosol absorption coefficient

Aerosol attenuation was measured online at seven wavelengths using dual-spot Aethalometers (AE33 model). For a selected set of 27 offline samples, we used the multi-wavelength absorption analyzer (MWAA; Massabò et al., 2013) to determine the total aerosol absorption at five wavelengths. The MWAA measures the transmission and backward scattering at two fixed angles, which takes into account the scattering contributions to attenuation. The filter absorbance measured by MWAA has been successfully validated against both a polar photometer and a multi-angle absorption photometer/MAAP (Massabò et al.,

2013) which, in turn, has been validated against numerous *in situ* methods. AE33-based attenuation measurements were then normalized by the MWAA-based absorption measurements to derive calibration coefficients, $C$ (Text S3.1) at different wavelengths using Eq. (S1). $C$ values, shown in Fig. A1, decreased from 2.5 at 370 nm to 2.3 at 880 nm on average, with day-to-day variability of ~15 % and no detectable systematic variation in time nor between sites. Average wavelength-dependent $C$ values were then applied to the loading-compensated AE33 attenuation coefficients, $b_{\text{ATN,AE33}}$, to obtain total aerosol

absorption coefficients, $b_{\text{abs,total}}$ (Eq. (S2)), for all samples at seven different wavelengths. The obtained range of $C$ values for the Swiss urban/rural dataset was close to, but statistically different from the widely used value of 2.14 derived for fresh, externally-mixed soot from fossil fuel emissions collected on quartz fiber filters using the AE31 model (Weingartner et al., 2003). In addition, selected samples were extracted in water, then in methanol and the resulting filters were sequentially measured by MWAA after extraction by each solvent to determine the wavelength dependence of the remaining aerosol (Sect.

3.4). The method is described in Text S3.1 and in Corbin et al. (2019), where it was applied for ship exhaust emissions.

### 2.1.4 UV-vis spectroscopy of methanol extracts

For all daily filter samples, $j$, absorbance spectra, $A_j(\lambda)$, of aerosol extracts in ultrapure water ($A_{\text{H2O}}$) or methanol ($A_{\text{MeOH}}$) were determined (Text S3.2; Eq. (S3)) at 280-600 nm using an ultraviolet-visible (UV-vis) spectrophotometer (Ocean Optics) coupled to a long-path detection cell (length $l$ = 50 cm). The UV-vis spectrophotometer and detailed analytical protocol for

absorbance measurements from extracts are described in our previous study for water (Moschos et al., 2018) and in the SI for MeOH. Methanol was selected based on the comparison of the absorbance obtained with five other solvents: water, acetonitrile, acetone, tetrahydrofuran and dichloromethane (Fig. S3). A dilution series of MeOH extracts at various concentrations showed linearity in the range of our extract concentrations and absorbance values below 500 nm (Fig. S4), indicating the applicability of Lambert-Beer's law. We assessed the water/methanol solvent effect on the extract absorbance (Mo et al., 2017) by

comparing $A_j(\lambda)$ for five ambient PM samples extracted in water and then diluted in methanol or in water in a 10:90 ratio. This ensured that identical organic samples were dissolved in the two different solvents, in order to examine the influence of the solvent on UV-vis spectra of the BrC samples. This is typically not considered in existing literature for BrC (Zhang et al., 2013; Kumar et al., 2018), although the solvent effects on molecular absorption have been extensively studied in organic chemistry (Reichardt, 2003). We observed in Fig. S5 higher absorbance in MeOH compared to water by a factor of 1.0-1.15

at wavelengths > 370 nm, consistent with the blue shift observed when more protic solvents (such as water) are used (Han et

al., 2003). We therefore scaled our $A_j(\lambda)$ in water to those in MeOH using an average MeOH/H$_2$O wavelength-dependent (for $\lambda < 470$ nm) absorbance ratio from these measurements. The results from (i) the comparison between different solvents, (ii) the dilution series of the MeOH extracts at various concentrations and (iii) the assessment of the water/MeOH solvent effect, indicate that the interactions of the BrC molecules with their matrix have little effect on their absorbance. This is in line with recent findings (McKay et al., 2017; Trofimova et al., 2019). We considered that $A_j(\lambda)$ is only related to the organic particulate matter soluble in water or MeOH, whereas insoluble (filtered out) black carbon, inorganic salts (confirmed based on Moschos et al. (2018)) and other species (e.g., organometallic complexes) are not expected to contribute significantly to the observed $A_j(\lambda)$. We also consider MeOH to extract the majority of absorbing organics (Fig. S6), as we will show below based on the MWAA and Sunset-OC mass measurements of aerosol filters upon extraction (Sect. 3.4).

The UV-vis provides mass absorption efficiency (MAE) spectra of an absorbing fraction, which are determined as the absorbance matrices $A$ normalized by the dissolved mass $M$ of this fraction:

$$\text{MAE}(\lambda) = \frac{A(\lambda)}{M} \tag{1}$$

The MAE can be related to the imaginary part of the refractive index, $k$, if the extractable (NR-PM) material density, $\rho_{solute}$, is known (1.5 g cm$^{-3}$ was assumed here):

$$k(\lambda) = \text{MAE}(\lambda) \cdot \frac{\rho_{\text{solute}} \cdot \lambda}{4\pi} \tag{2}$$

While $k$ is a material property that is independent of the geometry of the optical problem, MAE only relates to the absorption of the material in dilute solutions or in thin layers of the material in the pure form and cannot be directly applied to derive the absorption in the particle phase. We then use Mie calculations to determine the BrC mass absorption cross-section, MAC, in order to estimate the absorption of the organic fraction in the particle phase, as detailed in Sect. 2.3.

## 2.2 UV/Vis-PMF to infer methanol-soluble OA factor-specific MAE spectra

The UV/Vis-PMF statistical model aims at retrieving MAE spectra for each OA factor previously identified from the offline AMS/PMF analyses (Sect. 2.1.1). The methodology has been thoroughly described in Moschos et al. (2018), where it was applied to water extracts. Briefly, the model minimizes the residual difference between the observed $A_j(\lambda)$ (model input, in Mm$^{-1}$) and a reconstructed $A_j(\lambda)$. The latter is the product of the mass concentration time series ($M$) of each AMS/PMF OA factor (model constraints; in µg m$^{-3}$) and a matrix containing the factor-specific absorption efficiency spectra, MAE($\lambda$) (model output; in m$^2$ g$^{-1}$). The actual implementation was more complex than just described. Fixing the time series required an exchange of the roles played by time and wavelength and workarounds with respect to normalization (Moschos et al., 2018).

This approach provides MAE specific to an OA factor in hypothetical pure form, i.e., extracted in a solvent and externally mixed from other aerosol components. This approach may facilitate optical (e.g., radiative forcing) calculations starting from fundamental, intensive material properties.

Here, we applied the UV/Vis-PMF model to both methanol-soluble ($A_{MeOH}$) and water-insoluble ($A_{MeOH} - A_{H2O}$) absorbance matrices, upon correcting $A_{H2O}$ for the water/methanol solvent effect up to 470 nm (Fig. S5). The AMS/PMF factor selection (BBOA, WOOA, Other OA) for the UV/Vis-PMF model is described in Text S4 and Table S1. We obtained a range of factor-specific $k(\lambda)$ values by using the $M$ data from two different offline AMS/PMF solutions as model constraints and by reducing the spatial and temporal coverage of the input absorbance matrices (Text S4). Based on this approach, we obtained for each factor a median spectrum and the interquartile range (IQR) from the different model runs. We proceed below with the methanol-derived $k(\lambda)$ as representative of the total extractable BrC for optical calculations, assuming little structural changes of the chromophores upon dissolution (Mo et al., 2017; Lin et al., 2017; Dasari et al., 2019).

## 2.3 Determination of particulate BrC absorption

We used Mie calculations (Bohren and Huffman, 1998) to estimate the BrC absorption, $b_{abs,BrC-Mie,j}(\lambda)$, at four AE33 wavelengths (370 nm, 470 nm, 520 nm, and 590 nm) with Mie code programmed in the software package Igor Pro (WaveMetrics). The main inputs required for Mie calculations of light absorption coefficients are the particle size distribution (partially constrained here from SMPS measurements; Sect. 2.1.2) and the refractive index of the aerosol material in question. These calculations assume that the absorbing (BrC-containing) non-refractory particles are spherical with homogeneous internal composition in the particle phase (Sumlin et al., 2018; Li et al., 2020). In literature, a constant MAC/MAE ratio of 1.8-2.0 is often used to proceed from absorption measured in dilute solutions to absorption in the particle phase (Liu et al., 2013; Washenfelder et al., 2015; Shamjad et al., 2017; Zeng et al., 2020). Figure B1 shows the effect of particle size on the MAC/MAE ratios computed using Mie calculations for materials with different $k$ values, up to that of pure BC. A constant extractable aerosol density ($\rho_{solute}$) of 1.5 g cm$^{-3}$ and a wavelength-independent real part of the refractive index ($n$) of 1.5 (Lu et al., 2015) for all non-refractory (organic and inorganic) components were assumed, with respective values of 1.80 g cm$^{-3}$ and 1.95 for BC (Bond and Bergstrom, 2006; Cappa et al., 2012; Kim et al., 2015; Luo et al., 2018). No correction for the solvent refractive index was applied, considering literature values for $n_{MeOH}$ = 1.35 (Herráez and Belda, 2006) vs $n$ = 1.3-1.6 in particle solutions (Nakayama et al., 2013; Moise et al., 2015).

Figure B1 shows that the MAC to MAE ratios (MAC/MAE) are not constant and generally lower than used in literature. The calculated MAC values based on the Rayleigh regime for individual BrC molecules were 0.75 times the UV/Vis-based MAE values measured in solution, which is consistent with the range of 0.69-0.77 reported in previous studies (Sun et al., 2007; Liu et al., 2013; Nakayama et al., 2013). The MAC/MAE is highest for weakly absorbing particles in sizes in the Mie-regime, and drops again in the geometric regime. Unlike literature assumptions of a single MAC/MAE value, here we have assessed the

sensitivity of the MAC/MAE on the material $k$ and particle size. The methodology is detailed in Text S5. Briefly, we have considered two distinct size ranges: a smaller (120 nm) and larger size range (200-400 nm). The larger size is based on aerosol mass size distributions from Zurich (Fig. S7). The smaller range was considered to represent primary emissions (HOA + BBOA). It can be considered as a reasonable lower limit, determined based on the volume size distributions and the contribution of primary emissions to the total aerosol mass. In the absence of particle size distribution measurements for Magadino, we assume the same values for the larger and smaller size ranges as in Zurich. We do not have mixing state data; however, with treating NR-PM as homogeneous spheres externally mixed from BC, we assessed the sensitivity of resulting BrC absorption estimations ($b_{\mathrm{abs,BrC-Mie},j}(\lambda)$) to variations in mixing state and size of NR-PM components. Specifically, we have considered seven cases as detailed in Text S5, but here we only present the two extreme cases together with a central estimate obtained as *(min+max)/2*. The lowest estimate of $b_{\mathrm{abs,BrC-Mie},j}(\lambda)$ corresponds to BBOA at 120 nm, externally mixed with inorganic components; OOAs externally mixed from each other in the larger size range, externally mixed from inorganic components (case 6: average MAC/MAE for total OA ~1.3). The highest estimate corresponds to BBOA, OOA and inorganic components, all internally mixed in the larger size range (case 5: MAC/MAE ~1.6).

## 3 Results and discussion

### 3.1 Absorption characteristics of the bulk methanol-extracted OA

The WSOA (total OA) average concentration is 3.7 (5.4) and 6.0 (9.4) µg m$^{-3}$ in summer and winter respectively, accounting for ~66 % of the total OA. Figure 1 presents the methanol-extracted aerosol absorption characteristics. Figure 1a shows the water:methanol 370 nm absorbance ratio for the full PM$_{10}$ dataset, where the systematically lower average values observed at both sites in winter (Magadino: 0.73; Zurich: 0.74) compared to summer (Magadino: 0.82; Zurich: 0.88), indicate that MeOH extracts more absorbing matter than water when BBOA is prevalent. The coarse (PM$_{10}$–PM$_{2.5}$) aerosol fraction in Magadino during 2014 contributes on average 33 % to total OA mass (Fig. S8) but only ~5 % to absorbance in MeOH at 370 nm, lower than in another study (Chen et al., 2019). The two organic fractions that dominate the coarse mode OA are PBOA and SCOA, believed to be derived from vegetative detritus and non-exhaust car emissions, respectively (Bozzetti et al., 2016; Daellenbach et al., 2017; Vlachou et al., 2018). The low contribution of the coarse mode to the total absorbance suggests a negligible contribution of these two fractions to absorbance, as explained in Text S4 and shown in Table S1 and Fig. S8-S9.

***Insert Figure 1 here***

During winter (Oct-Mar), the bulk $k_{\mathrm{OA-MeOH,370nm}}$ is $0.032 \pm 0.013$ in Magadino (where BBOA is prevalent) and $0.024 \pm 0.007$ in Zurich (where WOOA is prevalent), 2-3 times higher on average than the summer average value of $0.010 \pm 0.006$ observed at both sites. The full dataset absorption Ångström exponent, AAE$_{370/\lambda}$ ($\lambda$ = 470/520/590 nm), of the MeOH extracts is 5.6 $\pm$

0.8. The average $k_{\text{OA-MeOH}}$ values of 0.020 at 370 nm and 0.003 at 590 nm (Fig. 1b) are in the same wide range as reported in previous field studies in Atlanta (Liu et al., 2013), Alabama (Washenfelder et al., 2015) and Beijing (Cheng et al., 2017).

## 3.2 Absorption properties of methanol-soluble OA factors derived from UV/Vis-PMF

We have applied UV/Vis-PMF (Sect. 2.2) to infer the absorption properties of methanol-soluble OA factors previously identified from offline AMS/PMF analyses (Fig. S2; Daellenbach et al., 2017; Vlachou et al., 2018). Figure S10 shows the spectrally-resolved relative contributions of the different OA fractions to the total measured absorbance in MeOH for both sites/seasons. In wintertime (Oct-Mar), the absorbance in Magadino is driven by two anthropogenic non-fossil (Vlachou et al., 2018) factors: primary BBOA and WOOA, the latter largely influenced by aged/transported biomass smoke. The Other OA

(Magadino: HOA, fOOA, SOOA; Zurich: HOA, COA, SOOA), processed and presented as one combined factor due to its weaker absorptivity, contributes less to absorbance (~20 %) despite its predominance in terms of mass, especially in summer (e.g., biogenic-dominated SOOA). The water-insoluble but methanol-soluble BBOA (WINS-BBOA; corrected for the water/methanol solvent effect) explains up to 30 % (averaged across 280-600 nm) of the total measured (explained and unexplained) absorbance in MeOH. The water-soluble/methanol-soluble BBOA absorbance ratio is between 0.45 and 0.77,

indicating that a non-negligible fraction of BrC in BBOA in this study is not water-soluble. This is consistent with the AMS/PMF recovery analysis estimating that two thirds of the total BBOA mass is water-soluble (Daellenbach et al., 2017; Vlachou et al., 2018), whereas the OOA components are almost fully water-soluble (Text S4).

Figure 2 shows the $k$ spectra of the different methanol-soluble OA source components derived from UV/Vis-PMF, where

(WINS-)BBOA have been corrected for their extraction efficiency in methanol (Text S4; Fig. S6). The source-specific BrC absorptivity spans 3 orders of magnitude and depends on the source and the wavelength, with $k_{\text{BBOA}} > k_{\text{WOOA}} >> k_{\text{otherOA}}$ at all wavelengths, whereas the bulk $k_{\text{OA}}$ values (Fig. 1b) are in-between the factor-specific ones. The estimated relative uncertainty for the retrieved factor-specific $k_{\text{BrC}}$ (Text S4 and Table S2) is ~15-20 % on average below 400 nm for the methanol-soluble factors and up to 40 % for water-insoluble BBOA. We do not observe significant differences in the factor-specific $k$ (and

contributions) between UV/Vis-PMF performed for different seasons/sites or using different AMS/PMF solutions as constraints. The water-insoluble but methanol-soluble BBOA fraction is more absorptive but mainly at shorter wavelengths (< 350 nm), and exhibits a steeper wavelength dependence of absorbance (AAE$_{\text{300-400nm}}$ ~6.1) compared to methanol-soluble BBOA (~4.9). The associated near-UV absorbing compounds may be less polar nitro-aromatics (Mohr et al., 2013), polyphenols (Lin et al., 2016) and polycyclic aromatic hydrocarbons with absorption peaks at 300-400 nm, or derivatives of

them (Chen et al., 2019). WOOA exhibits similar AAE in MeOH (~4.7) and water (Moschos et al., 2018), whereas Other OA has the highest AAE of ~7.1.

***Insert Figure 2 here***

The retrieved $k_{BBOA}$ (0.062 at 370 nm; Table S2) is comparable to those of lab-generated primary OA from biomass burning (Chen and Bond, 2010) and the combustion of other solid fuels (Lu et al., 2015), e.g., residential coal burning in China (Li et al., 2019), as well as to ambient aerosols heavily influenced by biomass burning emissions (Cappa et al., 2019; Yan et al., 2020) including firewood smoke and savannah fires (Kirchstetter et al., 2004; Lack et al., 2012). However, it is significantly lower than values found for flaming-phase wood burning emissions (Kumar et al., 2018), tar carbon from wildfires (Adler et al., 2019) and insoluble marine-engine exhaust tar balls (Corbin et al., 2019). $k_{WOOA}$ (0.030 at 370 nm; Table S2) is lower than that of fresh BBOA, and compares well with anthropogenic high-$NO_x$ toluene SOA (Liu et al., 2015a) in the near-UV and visible regions, as well as with a nitrate-associated OOA recently reported in Fresno (Cappa et al., 2019). The "Other OA" profile is similar to cooking- or fossil-fuel-influenced lab (Xie et al., 2017) and field (Cappa et al., 2012; Moschos et al., 2018) samples. The factor-specific $k$ (or MAE) spectra can be used in future studies to estimate at all wavelengths the particulate atmospheric BrC absorption attributable to different sources, at environments where these $k$ spectra apply.

### 3.3 Factor-specific OA contribution to particulate absorption

In Fig. 3, we present the calculated time-series of factor-specific BrC absorption in the particle phase, $b_{abs,BrC-Mie}$, for Zurich and Magadino at $\lambda = 370$ nm (data for 470 nm, 520 nm and 590 nm are shown in Fig. S11). In Magadino during winter, BBOA dominates the OA absorption with an average contribution of 84 % and daily values of $11 \pm 9$ Mm$^{-1}$ at 370 nm. The absorption during summer is lower, with WOOA (~0.5 Mm$^{-1}$) and Other OA (~0.7 Mm$^{-1}$) becoming more important in relative terms. In Zurich, all factors contribute significantly to total OA absorption at 370 nm (BBOA: 0.8 Mm$^{-1}$; WOOA: 1.1 Mm$^{-1}$; Other OA: 0.5 Mm$^{-1}$) with less distinct seasonal variability compared to Magadino. The average total extractable particulate BrC $AAE_{\lambda/590nm}$ for $\lambda = 370/470/520$ nm ($\pm$ day-to-day variability) is $6.0 \pm 0.2$ for Magadino winter, $5.2 \pm 0.3$ for Magadino summer and $5.5 \pm 0.2$ in Zurich.

***Insert Figure 3 here***

Similar to the finding of Shamjad et al. (2017), our mixed-source $MAC_{BrC}$ is not correlated with the bulk EC:BrC mass ratio (Fig. S12). This indicates that a lab-based parametrization linking the brownness of (primary) biomass burning emissions to their BC content (Saleh et al., 2014) may have limited applicability to ambient data. This is because also other primary absorbing OA than biomass burning, as well as secondary BrC, can contribute to absorption, and EC derives from multiple sources (both wood burning and traffic in this study).

### 3.4 Residual AAE of extracted filters indicating absence of tar-balls

We have conducted (sequential) extractions and MWAA measurements to determine the AAE of uncoated, pure BC and to examine the potential presence of (insoluble) tar carbon. The latter has been observed in atmospheric samples rich in fresh

biomass burning/wildfire smoke (China et al., 2013), and exhibits distinct absorption properties than those of BC and conventional BrC (Corbin et al., 2019). Figure 4a shows that the $AAE_{375-850nm}$ of $1.26 \pm 0.27$ for the untreated samples (black markers) measured with MWAA is larger than the AAE at longer wavelengths, consistent with the increasingly important BrC absorption at shorter wavelengths. Upon extraction in water (blue markers), the total AAE decreases to $1.04 \pm 0.18$ but remains higher than expected for BC, indicating the presence of water-insoluble absorbing non-BC components. This is consistent with the UV/Vis-PMF results attributing up to 30 % of the measured absorbance in MeOH to water-insoluble biomass burning emissions (Sect. 3.2). The subsequent extraction with MeOH (red markers) further decreases the AAE to $0.93 \pm 0.16$ for all wavelength combinations (850 nm used as reference), in line with the higher absorbance of MeOH extracts compared to water. The resulting AAE is consistent with that of pure BC across all wavelengths, indicating that the absorption at shorter wavelengths is not dominated by insoluble biomass burning tar carbon in this study, but by MeOH-soluble BrC. Previous laboratory work using the same technique observed that insoluble tar-balls, with AAE values comparable to those of soluble BrC, can dominate the BrC absorption from residual fuel combustion in marine ship engines (Corbin et al., 2019). The absence of tar-balls in our study may be related to the specific source of biomass-burning organics in our region, i.e., wood burning for residential heating in high-efficiency stoves rather than in low-efficiency wildfires, although more measurements of direct emissions would be needed to confirm this. Our $AAE_{bareBC}$ of $0.93 \pm 0.16$, obtained using filter samples from both sites/seasons thus covering all conditions, is within the predicted range for fresh, aged and/or compacted BC (Liu and Mishchenko, 2018; Liu et al., 2018) and experimental data from different emission types and BC ageing degree (Kirchstetter et al., 2004; Chung et al., 2012).

***Insert Figure 4 here***

Consistent with the aforementioned extractions, microscopic images of both treated and untreated samples also do not suggest a significant presence of tar-balls. While the only spherical particles observed in untreated Zurich samples were non-carbonaceous, either Fe-bearing or containing K, Mg, Ca, Al and S, in Magadino winter both bare BC and pseudo-spherical carbonaceous particles are observed, the latter disappearing after extraction with water (Fig. S13), consistent with the solubility of BrC determined above. We have also measured the extraction efficiency of biomass burning-dominated samples by determining the Sunset-OC mass on solvent-extracted and untreated filters. Based on this exercise, we find that at least 93 % of the Sunset-OC is MeOH-soluble (Fig. S6). These observations, together with the calculated $MAC_{bareBC}$ discussed in Sect. 3.5, provide a clear indication that there is no (significant) background bias (Adler et al., 2019) due to the potential presence of insoluble, refractory (tar) BrC with typically lower MAC than that of BC (Corbin et al., 2019).

**3.5 MAC$_{BC}$ and filter-based lensing effect**

We have used MWAA measurements to relate the AE33 attenuation to absorption coefficients (Sect. 2.1.3). This enabled us

to use the AE33 data quantitatively with no need to assume a range of possible calibration values (Kasthuriarachchi et al., 2020). Specifically, we use Eq. (S1) to calculate wavelength-specific calibration values representing the correction for multiple scattering by particles and filter fibers, and use these values to derive calibrated total absorption coefficients, $b_{abs,total}$ (Eq. (S2)) from AE33 data when MWAA measurements are not available. MAC$_{BC}$ is then estimated from the bulk measurements of absorption coefficients normalized by the EC mass (MAC$_{BC}$ = $b_{abs,total}$ / [EC]) at longer wavelengths (> 600 nm) where BC

is expected to dominate the absorption. Figure 4b displays the MAC$_{BC}$ against the contribution of fossil emissions to EC derived from $^{14}$C analysis (Vlachou et al., 2018), showing no relationship between the two variables. This finding is in line with Zotter et al. (2017) and Herich et al. (2011) inferring that MAC$_{BC}$ is largely source-independent, at least in Switzerland.

Figure 4c presents MAC$_{BC}$ as a function of a proxy for the BC coating thickness, i.e., the ratio between the combined

concentrations of major-SIA, OOA and BBOA and EC (NR-PM/EC; Table S4). While the variability in MAC$_{BC}$ is not driven by the EC sources (Fig. 4b), MAC$_{BC}$ increases linearly with NR-PM/EC < 33 consistently, unlike for other tested proxies (including the total OA mass, OA:EC, OOA:OA, OOA:EC or Sulfate:EC), indicating a filter-based BC lensing effect due to coating by multiple non-refractory components. The presence of coated BC particles is supported by the observation of compacted BC particles from SEM measurements (Fig. S13). We have examined the relationship between $C_{660nm}$ and the proxy

and found them to be independent, indicating that uncertainties in the Aethalometer calibration do not affect the resulting relationships between E$_{abs}$ and the proxy. While we attribute the filter-based (apparent) BC absorption enhancement to lensing, future studies should evaluate its potential dependence on chemical components that are externally mixed with BC, including tar-balls absorbing at longer wavelengths (Sect. 3.4), as well as calibration uncertainties and/or the deposited particle morphology.


MAC$_{bareBC}$ is then estimated from the intercept of the linear fit between MAC$_{BC}$ and NR-PM/EC (Fig. 4c). The obtained values of $6.3 \pm 0.3$ m$^2$ g$^{-1}$ at 660 nm and $4.5 \pm 0.2$ m$^2$ g$^{-1}$ at 880 nm show little variability among the different sites/seasons (Fig. S14) and are within the literature range. The value of MAC$_{bareBC,880nm}$ is consistent with that reported in a laboratory study of externally-mixed BrC/BC emissions from residential wood-burning ($4.7 \pm 0.3$ m$^2$ g$^{-1}$; Kumar et al., 2018), calculated as the

slope of a linear fit between MWAA-calibrated Aethalometer attenuation values vs. the Sunset-EC mass. Also, a review of ten recent direct measurements of absorption and mass with different in-situ instruments (Liu et al., 2020) concluded that uncoated (freshly-emitted) BC has a typical MAC of $6.6 \pm 0.6$ m$^2$ g$^{-1}$ at 660 nm (extrapolated from $8.0 \pm 0.7$ m$^2$ g$^{-1}$ at 550 nm using AAE$_{bareBC}$ = 1.0), which is within one standard deviation of the value recommended earlier (Bond and Bergstrom, 2006). The data scattering from the fitted line can be attributed to measurement errors, variability of BC size/physical properties and of

the internal mixing ratio of NR-PM coatings to BC (Liu et al., 2015b; Chakrabarty and Heinson, 2018), as well as to the location of BC within BC-containing particles (Adachi and Buseck, 2013).

The filter-based BC absorption enhancement factor ($E_{abs,BC,660nm}$) reaches a maximum of 2.0-2.5 at NR-PM/EC ~33 for both sites/seasons (Fig. S14), before plateauing, which is in agreement with multiple European rural background sites (Zanatta et

al., 2016) and with Mie theory calculations (Zanatta et al., 2018; Cappa et al., 2019). Detailed optical calculations (Wu et al., 2018) support our observed "lensing saturation" effect associated with the aforementioned plateau, although this effect may depend on the BC physical properties/morphology. Total coated-particle-to-BC mass ratios of up to 5 and 6-11 (total particle-core diameter ratios of ~1.4 and up to 2.2, respectively) are in general representative of partially-coated (not completely engulfed aggregates) and embedded/compacted/core-shell-like soot particles, respectively (Chakrabarty and Heinson, 2018;

Liu et al., 2020). The aforementioned mass ratio range for embedded (aged, heavily coated) soot is significantly lower (at least 3 times) than the proxy value of 30-40 where our maximum BC absorption enhancement is observed. This indicates that a small fraction of NR-PM is expected to be internally-mixed with EC leading to a corresponding absorption enhancement. Finally, considering the day-to-day variability in filter-based $E_{abs,BC}$ (Fig. 4c), the eBC mass concentration determined using the Aethalometer may not be accurate when using a fixed manufacturer $MAC_{BC}$ corresponding to an assumed constant $E_{abs,BC}$

over time.

Figure 5 summarizes filter-based $E_{abs,BC}$ values at longer wavelengths from recent studies, calculated as the average $MAC_{BC}$ from long time series normalized to a reference MAC for bare BC (Bond and Bergstrom, 2006). While it is not certain how close the filter-based BC lensing is to true lensing for airborne particles, given the tendency of the filter deposition process to

destroy part of the BC coatings, literature data of filter-based $E_{abs,BC}$ values are not significantly different from those based on in-situ measurements (Yuan et al., 2020 and references therein). The average filter-based $E_{abs,BC}$ at near-source/urban sites is slightly lower than the global average; there is no increasing trend towards remote sites far from any direct emission influence, possibly due to measurement errors, calibration uncertainties and/or the collapse of the BC "core" for high proxy values. Our full dataset filter-based $E_{abs,BC}$ at longer wavelengths (660 nm and 880 nm) of 1.36 ± 0.07 is in agreement with the global

literature average values.

***Insert Figure 5 here***

### 3.6 Optical closure of total absorption: black and brown carbon and filter-based lensing

In this section, we estimate the contributions of bare BC, BrC and filter-based lensing to the absorption at different wavelengths; the calculations are detailed in Appendix C. Briefly, we use the $MAC_{bareBC,660nm}$ and $AAE_{bareBC}$ to infer $MAC_{bareBC}$ at shorter wavelengths, which we use to determine bare BC absorption coefficients (in $Mm^{-1}$) for the daily samples by

multiplying with the Sunset-EC mass concentration. BrC absorption coefficients are determined based on Mie calculations in Sect. 2.3. We assign the difference between total measured minus the sum of calculated bare BC and BrC absorption coefficients to (filter-based) lensing. Results are displayed in Fig. 6 for Magadino winter, Magadino summer and Zurich yearly average.

Bare BC is found to contribute two thirds of the particulate absorption in the near-UV to infrared wavelength range. Particulate BrC contributes significantly to absorption in the near-UV region despite having lower absorptivity than BC, because the mass of BrC-containing OA is ~10 times greater (Fig. S12). The average contributions (± 1 SD corresponding to day-to-day variability) of particulate BrC to total measured absorption are 30 ± 14 % at 370 nm, 10 ± 6 % at 470 nm, 6 ± 4 % at 520 nm and 4 ± 2 % at 590 nm. BrC absorption from mostly primary biomass burning emissions is relatively more important at the rural site during winter, with a contribution of 45 ± 15 % at 370 nm. Absorption not explained by bare BC + BrC and therefore attributed to filter-based BC lensing contributes to a similar extent as BrC in the 370-880 nm range. The ratio of BrC/BC-lensing ranges between 0.25 ± 0.06 (urban/summer) and 0.65 ± 0.14 (rural winter).

***Insert Figure 6 here***

### 3.7 Observation of filter-based BC lensing suppression induced by BrC coatings

Figure 6 highlights a wavelength dependence of filter-based BC lensing (yellow area): compared to the average filter-based $E_{abs,BC}$ of ~1.4 at 660 nm and 880 nm (Fig. 5), the average filter-based $E_{abs,BC}$ at 370 nm (Eq. (C4)) is significantly lower, around 1.1 (Magadino winter: 1.05, Magadino summer: 1.07, Zurich: 1.13), while BC lensing is expected to be nearly wavelength-independent if the coating is non-absorbing. In all three cases, the total measured absorption at 370 nm (orange markers in Fig. 6) is lower than the calculated total absorption obtained by adding the absorption by BrC and absorption by BC including transparent shell lensing extrapolated from 660 nm (gray lines in Fig. 6). This reduction amounts to 14 % for Magadino winter, 18 % for Magadino summer and 13 % for Zurich. For all three cases, the reduction is moderate but remains higher than our best estimates of quantifiable uncertainties, as discussed in Appendix C. The significance of this effect beyond uncertainties is further discussed below for Fig. 8. The respective reduction in the filter-based BC lensing, $b_{abs,lensBC}$, at 370 nm is 87 %, 82 % and 63 % for the three cases, respectively. When averaged across 370-880 nm, this reduction (yellow area vs. the area defined by the gray line minus the brown area minus the black area)] becomes 16 %, 14 % and 11 %, respectively.

We now examine the relationship between the extent of filter-based lensing suppression and the contribution of BrC to the total absorption ($b_{abs,BrC-Mie}/b_{abs,total}$). For this, we compute the filter-based BC lensing ratio, which is a measure of the remaining lensing at a short wavelength compared to 660 nm, as defined in Eq. (3):

$$\text{Lensing ratio} = \frac{E_{\text{abs,BC},j}(\lambda)}{E_{\text{abs,BC},j,660\text{nm}}} = \frac{b_{\text{abs,total},j}(\lambda) - b_{\text{abs,BrC}-\text{Mie},j}(\lambda)}{b_{\text{abs,total},j,660\text{nm}}} \cdot \exp(\text{AAE}_{\text{bareBC}} \cdot ln(\lambda/660)) \tag{3}$$

In Eq. (3), $j$ represents a daily sample, $b_{abs,total}$ is the total aerosol absorption coefficient (Eq. (S2)), $b_{abs,BrC-Mie}$ is the Mie-predicted absorption coefficient for total extractable BrC (Eq. (S6)), $\text{AAE}_{\text{bareBC}}$ is the wavelength-dependence of the residual absorption after methanol extraction of the filters (Sect. 3.4), and $E_{\text{abs,BC}}$ is the filter-based BC absorption enhancement factor (Eq. (C4)). We plot the filter-based lensing ratio against $b_{\text{abs,BrC-Mie}}/b_{\text{abs,total}}$ in Fig. 7. Lensing suppression would result in lensing ratios below unity. Addition of externally mixed BrC to an aerosol would not alter the lensing ratio while increasing the $b_{\text{abs,BrC-Mie}}/b_{\text{abs,total}}$ thereby moving a data point horizontally to the right. Adding externally mixed BC to an aerosol would decrease the lensing suppression and the fraction of absorption by BrC in such a way that the data point would move along a straight line towards the point (0,1). Figure 7 shows that the filter-based (or apparent) BC lensing ratio is reduced with an increasing contribution of BrC to the total absorption at 370 nm, indicating that lensing suppression occurs when BrC makes a substantial contribution to the total absorption.

***Insert Figure 7 here***

While Fig. 6 discusses the extent of filter-based lensing suppression by BrC coatings for dataset subsets, we present in Fig. 8 a framework allowing to quantitatively examine absorption suppression for all samples. The framework is based on two of the most commonly used parameters to describe the aerosol absorption: the total aerosol $\text{AAE}_{\text{370-660nm}}$ and the contribution of BrC to the total absorption, $b_{\text{abs,BrC-Mie}}/b_{\text{abs,total}}$. Lensing suppression can be seen as an increase in the $b_{\text{abs,BrC-Mie}}/b_{\text{abs,total}}$ without a commensurate increase in the total aerosol $\text{AAE}_{\text{370-660nm}}$. The extent of suppression is shown as isopleth, calculated as the relative deviation of the measured total absorption from the predicted total absorption by adding the absorption by BrC and absorption by BC including transparent shell lensing extrapolated from 660 nm using $\text{AAE}_{\text{bareBC}} = 0.93$ (this is analogous to the difference between the orange markers and the gray line in Fig. 6). Uncertainty in the ratio $C_{\text{370nm}}/C_{\text{660nm}}$ used for the calibration of AE33 attenuation measurements translates into vertical error bars. Meanwhile, horizontal error bars include uncertainties in AE33 absolute calibration coefficients, $C_{\text{370nm}}$, and in the BrC absorption at 370 nm from Mie calculations, without contingency for errors associated with deviations from spherical homogeneous particles (see Appendix C). It can be clearly seen that the measured absorption at 370 nm (Fig. 8a) is systematically lower than the predicted absorption, which provides an indication that lensing suppression occurs. Over-prediction is observed for 84 % of data points. The $P_{25}$-$P_{75}$ ($P$: percentile) total absorption reduction range at 370 nm vs. 660 nm is 11-18 %. The contribution of BrC to the total absorption at 470 nm effectively remains around 3 times lower than at 370 nm. The differences between calculated and measured total absorption at 470 nm remain within our quantifiable uncertainties for the majority of data points (Fig. 8b), indicating that lensing suppression is trivial at this wavelength.

***Insert Figure 8 here***

Our results provide the first experimental indication for potential BC lensing suppression in atmospheric aerosols at 370 nm wavelength where BrC absorption is significant. The central estimate of filter-based $E_{abs,BC,370nm}$ is only ~80 % of the expected value if BrC were entirely externally mixed from BC. The inferred observed (filter-based) BC lensing suppression has been
predicted by theoretical calculations. Lack and Cappa (2010) demonstrated using Mie calculations that lensing can be suppressed with increasing shell absorptivity and/or thickness. The authors considered two cases. In the first case, they assumed a small BC core of 60 nm diameter coated with 40 nm of a material having a $k$ of ~0.05, which resulted in a BrC contribution to total absorption of 0.22. In this example, the $E_{abs,BC}$ reduction at 400 nm (vs. 750 nm) was predicted to be only ~5 %, which is at the low end of values found experimentally in our study for the same BrC contribution (for $0.15 < b_{abs,BrC-Mie}/b_{abs,total} <$
$0.30$, the $P_{25}$-$P_{75}$ filter-based $E_{abs,BC}$ reduction range at 370 nm vs. 660 nm is 13-23 %). In the second case, they modelled a large BC core of 300 nm diameter coated with 200 nm of the same material, which resulted in a similar BrC contribution to total absorption of 0.25. In this example, the $E_{abs,BC}$ reduction at 400 nm was predicted to be as high as ~30 %, which is much more important than values found here for the same BrC contribution. These calculations support our experimental findings of filter-based lensing suppression by BrC coatings. At the same time, the occurrence of this effect would require (i) large and
compact BC cores, (ii) significant fraction of the BrC internally mixed with BC particles, and (iii) high effective absorptivity of the brown coatings. Plausibility of such properties remains speculative as we do not have auxiliary measurements to further support or discard them. In addition, we note that Mie calculations used in Lack and Cappa (2010) and here are highly simplified and resulting estimates may differ from those using particle-resolved ensemble models (Fierce et al., 2020; Wu et al., 2020). In particular, core-shell models are less accurate for larger size parameters (when $D > \lambda$) and considering also that
particle-to-particle heterogeneity in composition is an important feature of lensing effects (Fierce et al., 2020). Therefore, the interpretation of our observations on the basis of core-shell models should be exercised with caution (Wu et al., 2018; Chakrabarty and Heinson, 2018) and additional controlled laboratory experiments and sophisticated modelling work would be needed to better constrain this effect. Finally, we note that our optical closure is limited in terms of interpretation of lensing effects, due to unquantifiable uncertainties potentially associated with filter sampling artifacts, possible chemical interactions
between airborne BrC molecules or with BC, and the use of simplified Mie calculations to obtain the particulate BrC absorption (Appendix C).

## 4 Summary and implications

This study attempted to provide a holistic approach to understand the spectrally-resolved absorption by atmospheric BrC and BC, using long time series of daily samples from filter-based measurements. We determined the wavelength-dependent MAE
in dilute bulk solutions for total methanol-extracted OA. The imaginary part of the refractive index of different OA fractions

was estimated by applying UV/Vis-PMF (Moschos et al., 2018). The resulting $k_{370nm}$ of methanol-soluble BBOA, WOOA and Other OA were 0.06, 0.03 and 0.006, respectively. We attributed the totality of the NR-PM absorption at shorter wavelengths to methanol-extractable BrC, and demonstrated that the oxygenated OA component linked to anthropogenic secondary OA can be as important as primary biomass smoke for BrC absorption, especially at the urban background site. As for BC, our results suggest that the MAC of bare BC particles is independent of its source with a MAC of ~6.3 m$^2$ g$^{-1}$ at 660 nm and an AAE of 0.93 ± 0.16. Our observations provide clear evidence for absorption enhancement-associated non-refractory coatings, which increase the MAC of BC by a factor of ~1.4 on average at longer wavelengths through the so-called lensing effect. This enhancement factor falls within the range of previously reported values, 1.5 ± 0.3, from studies all over the globe applying filter-based techniques.

We have performed optical closure between measured and calculated total absorption at near-UV wavelengths, where the latter is obtained by addition of the absorption by BrC constrained with UV-vis absorbance spectroscopy of filter extracts combined with Mie calculations, and the absorption by BC including a transparent shell lensing constrained by the total absorption at 660 nm combined with extrapolation to near-UV wavelengths. Based on this closure, we provide first experimental indication of lensing suppression in real-world samples. The effect is moderate and remains restricted to shorter wavelengths and additional controlled laboratory experiments and sophisticated modeling work would be needed in future studies to better constrain the lensing suppression effect, which we derive here based on simplified Mie-calculations.

While the optical absorption closure approach presented here involves multiple assumptions and simplified concepts, our results provide useful experimental insights into understanding BrC/BC interactions and total atmospheric aerosol absorption, as well as uncertainties of filter-based measurements for quantifying BrC absorption. If lensing suppression occurs due to internal mixing of BC and BrC as is apparently the case for many samples in our study, then the additional absorption by BrC would be partially compensated by a concurrent lensing factor reduction. As a consequence, very different results for the BrC contribution to total absorption may be obtained by following either of the two approaches: i) estimation of the BrC contribution from extracts and Mie calculations (as considered in this study) or ii) estimation of the BrC contribution from the total measured absorption minus the absorption by BC + transparent shell lensing. We attempted to calculate the BrC absorption by subtracting the total BC contribution from the total calibrated absorption (approach ii), using an AAE$_{BC,\lambda/660}$ of 0.93 and considering that only BC contributes to light absorption at 660 nm. This approach led to considerably lower estimates of BrC absorption, i.e., around 64 % less on average at 370 nm compared to the average $b_{abs,BrC-Mie,370nm}$ that we obtained with approach (i). Therefore, if lensing suppression occurs, the $b_{abs,BrC}$ at shorter wavelengths obtained from the classical additive BC/BrC model (e.g., Zhang et al., 2020) could be biased low, or the total absorption calculated with the additive BC/BrC model and $b_{abs,BrC}$ constrained through material properties and optical models neglecting mixing effects could be biased high.

Our study allows a better understanding of the interactions between BrC, BC and non-absorbing PM and their influence on the optical absorption profile of the aerosol, often described by the AAE. The AAE from Aethalometer measurements is often used to distinguish between the eBC from wood burning emissions (high AAE) and from traffic emissions (low AAE). We show that the BrC fraction can vary without a concurrent variation of AAE between 370 nm and red to near-infrared wavelengths, potentially as a consequence of lensing suppression effects for internal mixtures of BC and BrC. This suggests that 470 nm may be a better choice than 370 nm for the short wavelength in the Aethalometer SA model, which is often applied for eBC source apportionment. When BBOA contributes more than 30-40 % to OA mass (e.g., Magadino winter), the absorption of BBOA will dominate, but when the BBOA contribution to OA mass is close to 10 % (e.g., Magadino summer or Zurich), the less absorptive OOA can be expected to dominate the OA absorption in the absence of other highly absorptive primary BrC sources. Therefore, our results suggest that aging produces light-absorbing OOA coatings, which may significantly contribute to BrC absorption. However, due to lensing at longer wavelengths and lensing suppression at shorter wavelengths, this high contribution may not be visible in the total aerosol AAE values, which cluster around ~1-1.1.

Finally, we estimate the fraction of solar radiation absorbed by carbonaceous species in the 300-900 nm range, for our dataset covering a wide range of conditions in the atmosphere. In Fig. 9, we compare the overall optical significance of atmospheric BrC, bare BC and BC lensing. This is based on the calculation of the rate of energy transfer per volume due to light absorption by total carbonaceous aerosol, $RET_{tot}$, and the fractional RET, $W$, due to light absorption by a fraction $X$, relative to that due to the total carbonaceous aerosol, $RET_X / RET_{tot}$. Calculations are shown in Appendix D.

***Insert Figure 9 here***

Figure 9 shows that the dominant contribution to solar radiation absorption stems from BC alone (around two thirds), amplified by the lensing effect (min to max IQR: 3-34 %). Data with negative lensing contributions are not statistically different from zero and are mainly related to errors in the $MAC_{bareBC}$ determination (Sect. 3.5). The average contributions by BrC vary between 8 ± 3 % (summer at both sites and Zurich) and 17 ± 7 % (Magadino winter) with a maximum of ~40 %. The $P_5$-$P_{95}$ of the full-dataset $W_{BrC}$ considering the overall uncertainties is 3-26 %. Therefore, extractable particulate atmospheric BrC is an optically relevant carbonaceous component especially at places affected by intense biomass/residential burning activity. Also, BrC can become the predominant absorbing species in the UV region (< 400 nm, ~7 % of sun's energy), with calculated average contributions between 50 ± 16 % (Magadino winter) and 29 ± 11 % (rest of the data), and thus can potentially affect together with BC (Kirchstetter and Thatcher, 2012) photochemical processes/reactions occurring in the troposphere.

# Appendices

## A. AE33 calibration coefficients

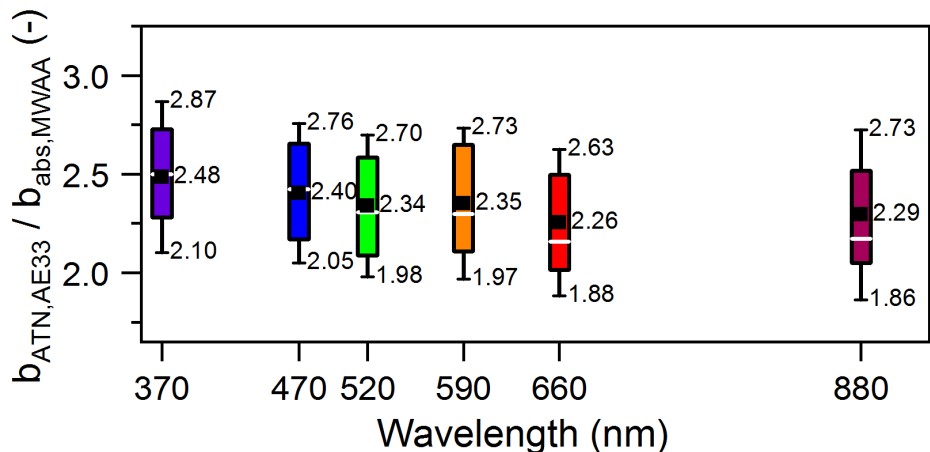

**Figure A1.** Wavelength-specific AE33 calibration coefficients, $C(\lambda)$, upon comparison of the loading-compensated AE33 attenuation with the MWAA absorption coefficient (Text 2.1.3; Text S3.1) for 27 quartz fiber filter samples from Magadino 2013 ($PM_{10}$) & Magadino 2014/Zurich 2013 ($PM_{2.5}$) [white line/box: median/IQR; black squares & whiskers: mean & 1 SD (labels)].

## B. MAC/MAE at 370 nm

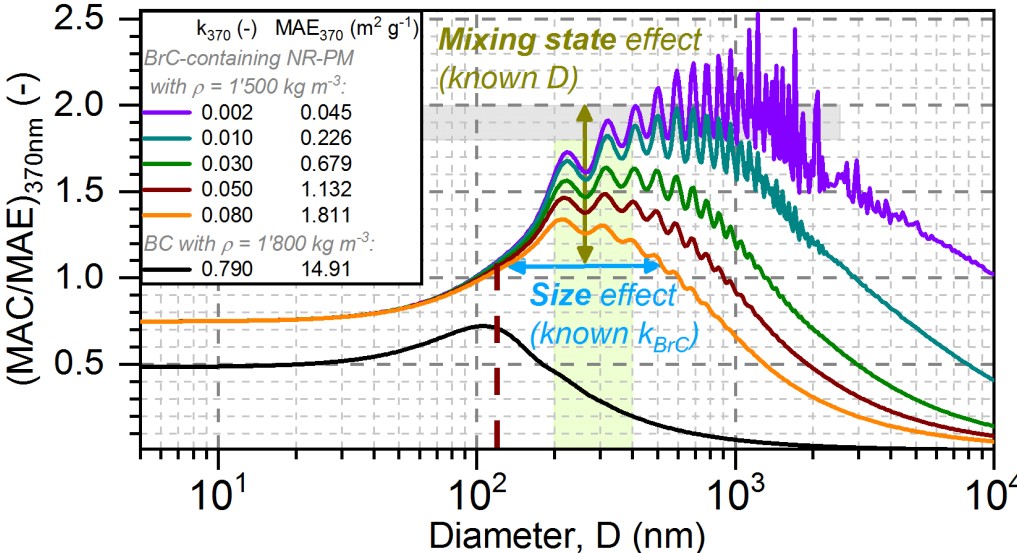

**Figure B1.** Effect of particle size on the MAC/MAE ratios at 370 nm for BrC-containing NR-PM particles with absorptivity up to that of pure BC. The different curve colours correspond to approximate absorptivity of individual methanol-extracted fractions determined by UV/Vis-PMF (orange: WINS-BBOA; brown: methanol-soluble BBOA; green: WOOA; dark cyan/violet: range of "Other OA" absorptivity). The light green box and thick dashed brown line indicate the larger size range mode and the lower limit, respectively, considered for Mie calculations. The light gray box indicates the MAC/MAE range of 1.8-2.0 previously reported in literature.

### C. Contribution of different fractions to total absorption and related uncertainties

We detail here the methodology we followed to calculate the contribution of different fractions to total absorption measured by AE33. We used as direct input the estimated BrC absorption using Mie-calculations (Sect. 2.3). We estimated the bare BC absorption at $\lambda \geq 660$ nm, $b_{abs,bareBC,j}(\lambda \geq 660$ nm), using EC time-resolved concentrations, $[EC]_j$, and the MAC of bare BC at $\lambda \geq 660$ nm, $MAC_{bareBC}(\lambda)$, based on Eq. (C1). The $MAC_{bareBC}(\lambda)$ is estimated based on the extrapolation of $MAC_{BC}(\lambda)$ at NR-PM/EC ratios = 0 (Sect. 3.5). The EC relative error is defined as: $\sqrt{(0.03 \cdot OC)^2 + (0.03 \cdot EC)^2}$.

$$b_{abs,bareBC,j}(\lambda \geq 660\ nm) = MAC_{bareBC}(\lambda) \cdot [EC]_j \tag{C1}$$

The bare BC absorption at < 660 nm was estimated using the AAE measured using the MWAA upon methanol extraction, $AAE_{bareBC}$ (Sect. 3.4), to extrapolate the bare BC absorption from 660 nm, $b_{abs,bareBC,j}(\lambda = 660$ nm).

$$b_{abs,bareBC,j}(\lambda < 660\ nm) = b_{abs,bareBC,j,660nm} \cdot [\exp(AAE_{bareBC} \cdot ln(660/\lambda))] \tag{C2}$$

BC lensing was attributed to the remaining absorption by subtracting the calculated absorption of bare BC and BrC from the total calibrated AE33 absorption:

$$b_{abs,lensBC,j}(\lambda) = b_{abs,total,j}(\lambda) - b_{abs,BrC-Mie,j}(\lambda) - b_{abs,bareBC,j}(\lambda) \tag{C3}$$

The $b_{abs,lensBC,j}(\lambda)$ term represents the transparent shell lensing at longer wavelengths, or the residual (BC) lensing at shorter wavelengths. We then define the time-resolved BC absorption enhancement factor at all wavelengths as follows:

$$E_{abs,BC,j}(\lambda) = \frac{b_{abs,total,j}(\lambda) - b_{abs,BrC-Mie,j}(\lambda)}{b_{abs,bareBC,j}(\lambda)} \tag{C4}$$

We present below the quantifiable uncertainties used to produce the error estimates in Fig. 6 and 8 and discuss non-quantifiable uncertainties. We note that the uncertainties presented are lowest estimates.

**Quantifiable uncertainties.**

Figure 6:
- Gray lines: uncertainties at 370 nm include the standard error (SE, 1 standard deviation, SD and the total number of points considered, $N$: SE=SD/sqrt($N$-1)) of $C_{370}/C_{660}$ and $AAE_{bareBC}$, and the uncertainty on $b_{abs,BrC-Mie}$ estimation. The latter includes the uncertainty in the bulk BrC MAC/MAE ratio and the SE of the UV-vis measurements.
- Orange markers (blue error bars): SE on the $C$ values calculated from Fig. A1.
- Black line (violet error bars): uncertainties on $MAC_{bareBC}$ (determined from the confidence interval in Fig. S14) and SE of $AAE_{bareBC}$, and EC mass concentration, propagated through Eq. (C1)-(C2).
- Brown line (red error bars): uncertainties on $b_{abs,BrC-Mie}$ estimation (including the uncertainty in the bulk BrC MAC/MAE ratio and the SE of the UV-vis measurement).
- Yellow line (green error bars): propagated uncertainties using Eq. (C3).

Figure 8:
- Y-axes: percentage uncertainties from the 1 SD of the $C_{370}/C_{660}$ or the $C_{470}/C_{660}$ ratio. The average ($\pm$ 1 SD) $C_{370}/C_{660}$ and $C_{470}/C_{660}$ ratios are $1.11 \pm 0.14$ and $1.07 \pm 0.09$, respectively.
- X-axes: propagated uncertainties from the $b_{abs,BrC-Mie}$ estimation as discussed above and the 1 SD of the $C$ values calculated from Fig. A1.

**Non-quantifiable uncertainties.**

We list below potential errors that could not be quantified but could affect our results:

- Sampling adsorption/desorption artefacts associated with mixing results from real-time vs. offline filter based measurements.
- Extrapolation of behavior of BC core with hypothetical transparent shell from 660 nm to 370 nm with constant AAE. We consider this to be a reasonable assumption (sensitivity is illustrated in Fig. 8) and therefore a minor source of uncertainties.
- Uncertainties related to potential chemical interactions between airborne BrC molecules in their concentrated media or with BC, and their effect on the BC/BrC contributions to the mixture absorption (Andersson, 2017).

- Uncertainties related to using an oversimplified optical model, based on Mie calculations, to obtain BrC absorption. This is most likely the greatest source of uncertainty that has not been considered in our analysis. Therefore, while our results show the first evidence of lensing suppression at shorter wavelengths, additional controlled laboratory experiments and sophisticated modelling work would be needed to better constrain this effect.

## D. Optical significance of different absorbing fractions in the atmosphere

The optical significance of particulate extractable BrC (Sect. 2.3) was assessed using the solar radiation spectrum (actinic flux at the Earth's surface) and an approach similar to that of Kumar et al. (2018). The wavelength-dependent solar emission flux, $S(\lambda)$ (in W m$^{-2}$ nm$^{-1}$), is given by Levinson et al. (2010) in which data obtained from the clear sky Air Mass 1 Global Horizontal

(AM1GH) solar irradiance model. We calculated for each sample, $j$, the rate of energy transfer (RET) per volume (in W m$^{-3}$) for each absorbing fraction $X$, i.e., particulate BrC, bare BC and BC lensing (where $RET_{lensBC} = RET_{tot} - RET_{bareBC} - RET_{BrC}$), and the fractional $RET$, $W$, to the air mass due to the absorption by each fraction $X$ compared to that of total carbonaceous aerosol, according to Eq. (D1):

$$W_{X,j} = \left. \mathrm{RET}_{X,j} \middle/ \mathrm{RET}_{tot,j} = \left. \int_{300}^{900} [b_{\mathrm{abs},X,j}(\lambda) \bullet S(\lambda)] \bullet \mathrm{d}\lambda \middle/ \int_{300}^{900} [b_{\mathrm{abs},total,j}(\lambda) \bullet S(\lambda)] \bullet \mathrm{d}\lambda \right. \right. \tag{D1}$$

We used an exponential decay model fit (5 nm resolution) for the $b_{\mathrm{abs},j}$ of a fraction obtained at six AE33 wavelengths (370-880 nm), because the wavelength range and increments of the Aethalometer, the UV-vis spectrophotometer and the solar spectrum datasets were different. Absorptions in the range 300-370 nm were obtained by extrapolating with a constant AAE based on observed spectral dependence of each fraction and are therefore less accurate. The range 280-300 nm was not considered in our calculations since virtually no radiation below 300 nm can reach the Earth's surface, an approximation that

could cause greater bias at high altitudes, where such, UV radiation is not fully attenuated by the atmosphere above.

## E. Abbreviations

AAE, absorption Ångström exponent; AE33, Aethalometer (33 model); AMS, aerosol mass spectrometry; BBOA, biomass burning organic aerosol; (e)BC, (equivalent) black carbon; BrC, brown carbon; EC, elemental carbon; IQR, interquartile range;

MAC, mass absorption cross-section; MAE, mass absorption efficiency; MeOH, methanol; MWAA, multi-wavelength absorption analyzer; NR-PM, non-refractory particulate matter; PMF, positive matrix factorization; RET, rate of energy transfer; SD, standard deviation; SE: standard error; UV-vis, ultraviolet-visible spectroscopy; WINS, water-insoluble (but methanol-soluble); WOOA, winter-oxygenated organic aerosol.

## F. Glossary

| Symbol | Unit | Definition | Equation |
|---|---|---|---|
| AAE | - | absorption Ångström exponent | $\dfrac{-ln(b_{\mathrm{abs},\lambda 1}/b_{\mathrm{abs},\lambda 2})}{ln(\lambda 1/\lambda 2)}$ |
| $A$ | Mm$^{-1}$ | time series of extracted aerosol absorption coefficient (absorbance) spectra used as input in UV/Vis-PMF | $A_j(\lambda)$    (S3) |
| $b_{\mathrm{abs}}$ | Mm$^{-1}$ | absorption coefficient of aerosol | - |
| $b_{\mathrm{abs,bareBC}}$ | Mm$^{-1}$ | absorption coefficient of bare (uncoated) black carbon | C1, C2 |
| $b_{\mathrm{abs}\,\mathrm{Mie}}^{\mathrm{BrC}-p}$ | Mm$^{-1}$ | Mie-based particle-type-specific BrC absorption coefficient | S5 |
| $b_{\mathrm{abs,BrC-Mie}}$ | Mm$^{-1}$ | average total extractable particulate BrC absorption | S6 |

| | | | |
|---|---|---|---|
| $b_{abs,lensBC}$ | Mm$^{-1}$ | Filter-based BC absorption enhancement (lensing) | C3 |
| $b_{abs,MWAA}$ | Mm$^{-1}$ | total aerosol absorption coefficient measured and reported by MWAA | - |
| $b_{abs,total}$ | Mm$^{-1}$ | MWAA-calibrated AE33 total aerosol absorption coefficient | S2 |
| $b_{ATN,AE33}$ | Mm-1 | filter attenuation measured and recorded by AE33, prior to calibration by MWAA | - |
| $C(\lambda)$ | - | wavelength-specific AE33 calibration coefficients to account for multiple scattering | S1 |
| $D$ | nm | particle diameter | - |
| $E_{abs,BC}$ | - | Filter-based BC absorption enhancement factor | C4 |
| $\dfrac{E_{abs,BC,j}(\lambda)}{E_{abs,BC,j,660nm}}$ | - | Lensing ratio (filter-based) | 3 |
| $f$ | - | mass fraction an aerosol component to the mixed NR-PM components in different particle mixtures considered for Mie calculations | - |
| $j$ | - | index for the j$^{th}$ number of 310 filter samples | - |
| $k$ | - | imaginary part of the refractive index | 2 |
| $k_{mix}$ | - | effective particle imaginary part of the refractive index (absorptivity) used as input in Mie calculations | S4 |
| $M$ | µg m$^{-3}$ | mass concentration time series of an AMS/PMF factor used (normalized) as constraint in UV/Vis-PMF | - |
| MAC$_{BC}$ | m$^2$ g$^{-1}$ | BC mass absorption cross-section | $\dfrac{b_{abs,BC}}{EC}$ |
| MAE | m$^2$ g$^{-1}$ | bulk solution mass absorption efficiency | 1 |
| $n$ | - | real part of the refractive index | - |
| RET$_X$ | W m$^{-3}$ | rate of energy transfer per volume due to light absorption by a fraction $X$ | D1 |
| $S(\lambda)$ | W m$^{-2}$ nm$^{-1}$ | solar irradiance spectrum | - |
| $X$ | - | placeholder for aerosol component (e.g., AMS/PMF factor, inorganics, total OA) | - |
| $W_X$ | - | fractional energy transfer due to light absorption by a fraction $X$ relative to that due to the total carbonaceous aerosol absorption | RET$_X$ / RET$_{tot}$ |
| $\lambda$ | nm | wavelength of light | - |
| $\rho_{solute}$ | g cm$^{-3}$ | density of extractable non-refractory particulate matter | - |

**Data availability**

Data will be made available on Zenodo upon acceptance.

**Author contributions**

VM and IeH conceptualized and formulated the study. VM performed the UV-vis measurements of methanol and water extracts. DM performed MWAA measurements before and after extractions. CC performed/discussed the FE-SEM/EDX analyses. SS and AV performed the $^{14}C$ measurements. KRD and AV performed the AMS/PMF analyses. VM performed the UV/Vis-PMF analyses. RLM, MGB, VM and IeH put the theoretical framework for optical interpretations and performed the Mie calculations. VM, IeH, MGB, RLM, and JCC interpreted the results. VM prepared the figures and wrote the manuscript. IeH, ASHP, UB & MGB supported and supervised the research. All authors reviewed and commented on the paper.

**Competing interests**

The authors declare no competing financial interest.

**Acknowledgements**

We acknowledge Zsófia Jurányi for the Mie code programmed in the software package Igor Pro and Jihwan Choi and Katherine Rowlands for experimental assistance. The authors thank Denise Verhoeven and Prof. Julia Schmale for administrative support.

**Financial support**

This work has been supported by funding from the European Union's Horizon 2020 Framework Programme via the ERA-PLANET project iCUPE (grant agreement no. 689443), the Swiss Federal Office for the Environment (FOEN), and the Swiss State Secretariat for Education, Research and Innovation (SERI; contract no. 15.0159-1). The opinions expressed and arguments employed herein do not necessarily reflect the official views of the Swiss Government or the authors' affiliated institutions. VM acknowledges the A.G. Leventis Foundation for a graduate student Educational Grant.

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

**Main figures**

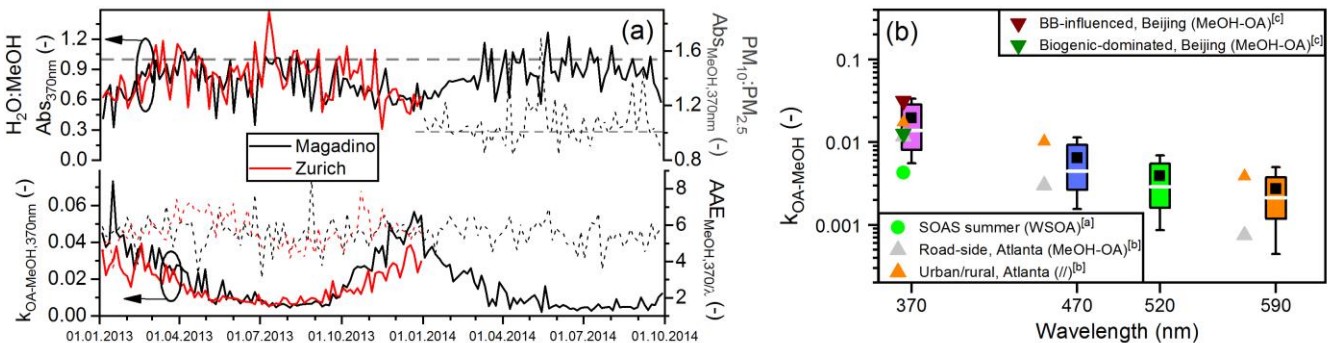

Figure 1: a) Time series at 370 nm of the water-to-methanol absorbance ($A$, or $Abs$) ratio (not corrected here for the solvent effect) and the $PM_{10}$-to-$PM_{2.5}$ absorbance ratio in methanol (horizontal dashed lines indicate ratios of 1.0), as well as the imaginary part of refractive index ($k$) of the bulk methanol-extracted organics (Eq. (2)) and the $AAE_{MeOH,370/\lambda}$ ($\lambda$=470/520/590 nm). The OA mass from AMS/PMF Solution 2 was used for all 2013 data, for a direct comparison between Magadino and Zurich. b) Box plots of the bulk mixed-source UV/Vis-based $k_{OA}$ (total extractable OA, homogeneous and internally mixed without any inorganics or BC present) at four AE33 wavelengths, for all 245 $PM_{10}$ samples measured from Magadino and Zurich [white line/box: median/IQR; black squares & whiskers: mean & 1 standard deviation (SD)], compared to selected literature values for bulk extracted OA [a: Washenfelder et al. (2015), b: Liu et al. (2013), c: Cheng et al. (2017); SOAS: "The Southern Oxidant and Aerosol Study"].

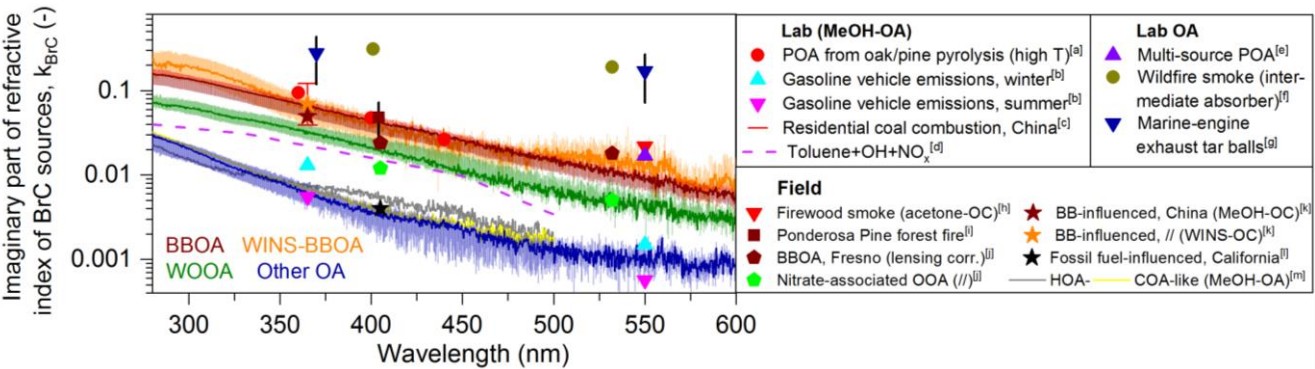

Figure 2: Spectrally-resolved median $k$ values (MAE in Fig. S10) of the different methanol-soluble (water-soluble: Moschos et al., 2018) offline AMS/PMF-based OA source components [(WINS-)BBOA corrected for extraction efficiency in MeOH; WINS-BBOA corrected for the water/methanol solvent effect; Other OA: HOA+COA+SOOA+fOOA], together with the IQR from different UV/Vis-PMF sensitivity runs (Text S4 & Table S2). The retrieved $k_{BrC}$ are compared to those of previous laboratory and field studies focusing on methanol-extractable or total organics [a: Chen & Bond (2010), b: Xie et al. (2017), c: Li et al. (2019), d: Liu et al. (2015), e: Lu et al. (2015), f: Adler et al. (2019), g: Corbin et al. (2019), h: Kirchstetter et al. (2004), i: Lack et al. (2012), j: Cappa et al. (2019), k: Yan et al. (2020), l: Cappa et al. (2012), m: Moschos et al. (2018)].

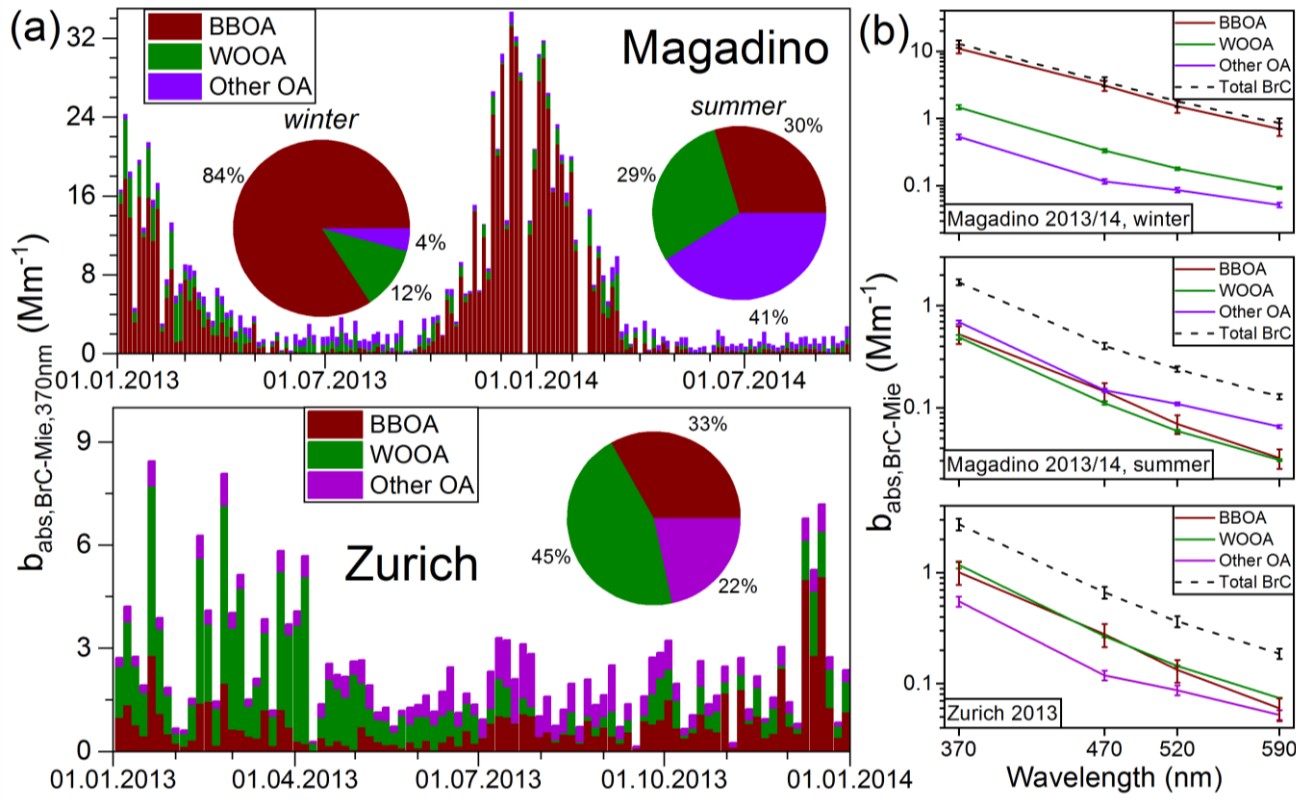

**Figure 3: a)** Time series of the cumulative BBOA, WOOA and Other OA (Magadino: HOA+fOOA+SOOA; Zurich: HOA+COA+SOOA) average (from 7 cases; Text S5) factor contributions to UV/Vis-PMF-based Mie-predicted particulate BrC absorption at 370 nm for Magadino and Zurich. The insets show the seasonal/yearly average (summer: Apr-Sep) factor relative contributions to $b_{abs,BrC-Mie,370nm}$. Note the different temporal coverage and the different range

of $b_{abs}$ values between the upper and bottom panel. **b)** Factor-specific particulate BrC absorption coefficient at four AE33 wavelengths for three distinct cases. Error bars correspond to the MAC/MAE uncertainty from the 7 cases (Text S5).

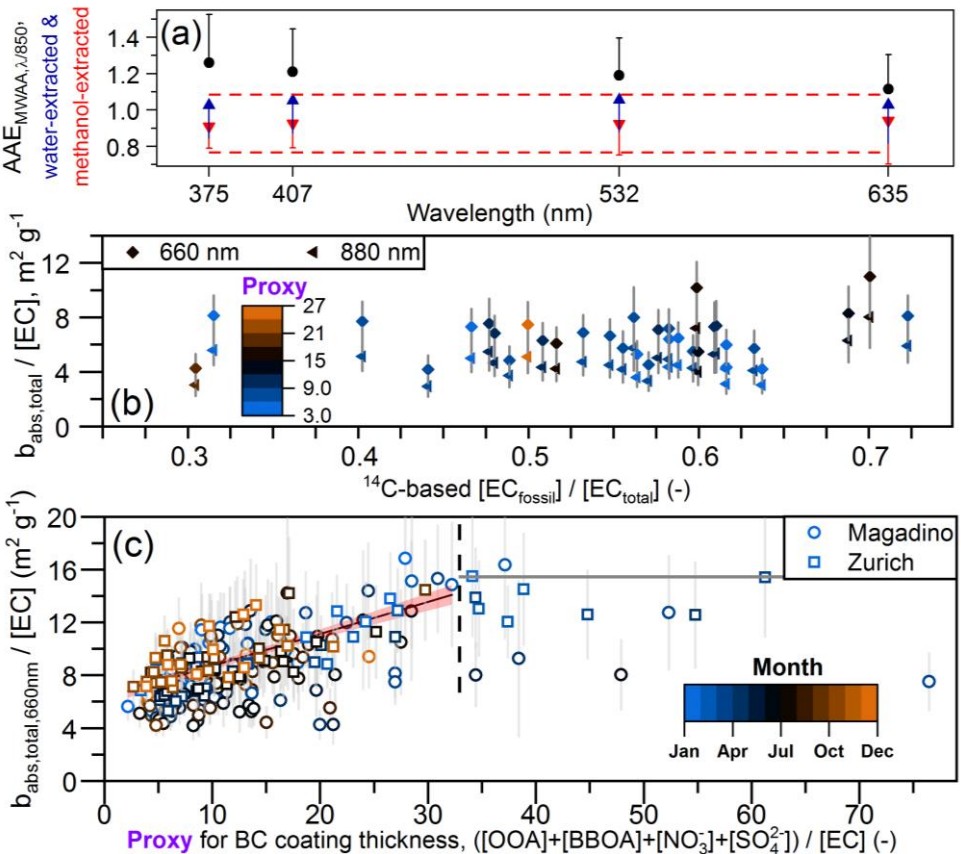

**Figure 4: a)** AAE of bare BC for nine different samples from both sites/seasons, measured with MWAA upon water and (sequential) methanol extraction for the removal of BrC/coatings, using 850 nm as reference wavelength. The error bars indicate 1 SD from all samples. The red dashed lines are "a guide for the eye" for the obtained range (± 1 SD) from all wavelengths upon methanol extraction. **b)** Assessment of possible source-related $MAC_{BC}$ variability at 660 and 880 nm [color-coded with the proxy; see $x$-axis in (c)], where extractable BrC does not contribute to particulate absorption, for year-long Magadino filter samples measured for $^{14}C$ in the EC fraction, relevant also for the application of the Aethalometer SA model. $R^2$ for 660 nm is 0.11 ($p$ = 0.06) and 0.16 for 880 nm ($p$ = 0.02). **c)** Calculation of $MAC_{bareBC}$ (Fig. S14), as the intercept of uncertainty-weighted linear regression of $MAC_{BC,660nm}$ vs. a proxy for the BC coating thickness, for proxy < 33. Intercept: 6.3 ± 0.3 $m^2$ $g^{-1}$; slope: 0.242 ± 0.019 $m^2$ $g^{-1}$; Pearson's $r$ = 0.64. The gray line for larger proxy values indicates the upper limit (+ 1 SD) of the observed $MAC_{BC}$ in this (lensing saturation) range.

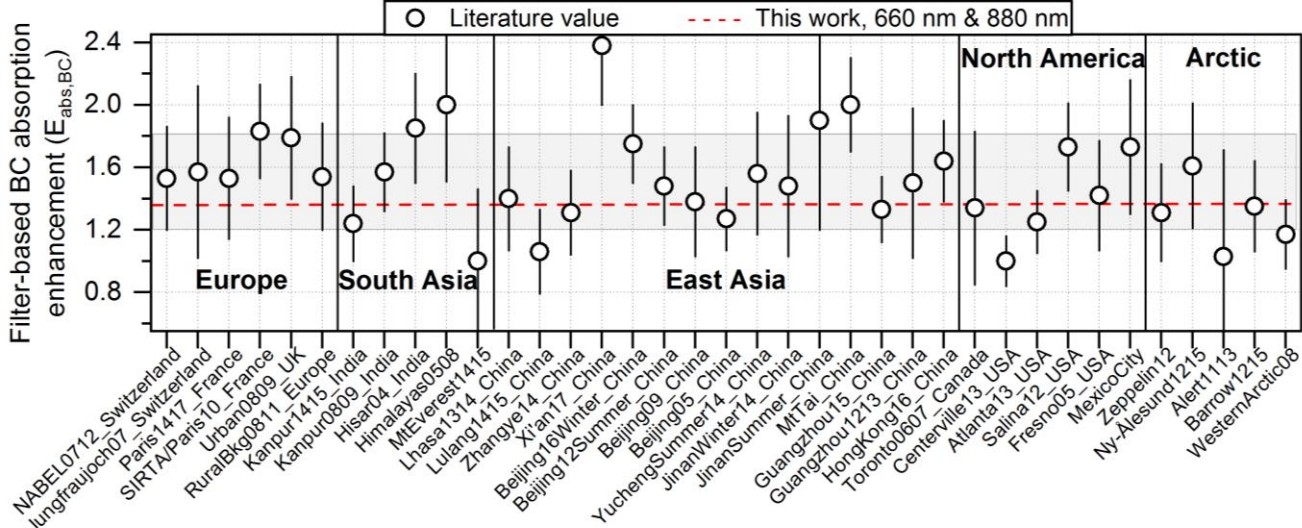

**Figure 5: Summary of campaign-average (error bars: ± 1 SD) filter-based BC absorption enhancement factor (Eq. (C4)) at longer wavelengths (> 600 nm) reported or inferred, using/extrapolating an observational reference value for uncoated BC MAC (Bond and Bergstrom, 2006), using the most common instruments (Aethalometer, PSAP, MAAP, Sunset/DRI, OCEC analyzer) at various locations/years. The gray-shaded area shows the global average within 1 SD (1.5 ± 0.3). The red horizontal dashed line denotes the full dataset average filter-based $E_{abs,BC}$ obtained in this study > 600 nm (1.36) where BrC does not contribute to absorption (Fig. 1-3).**

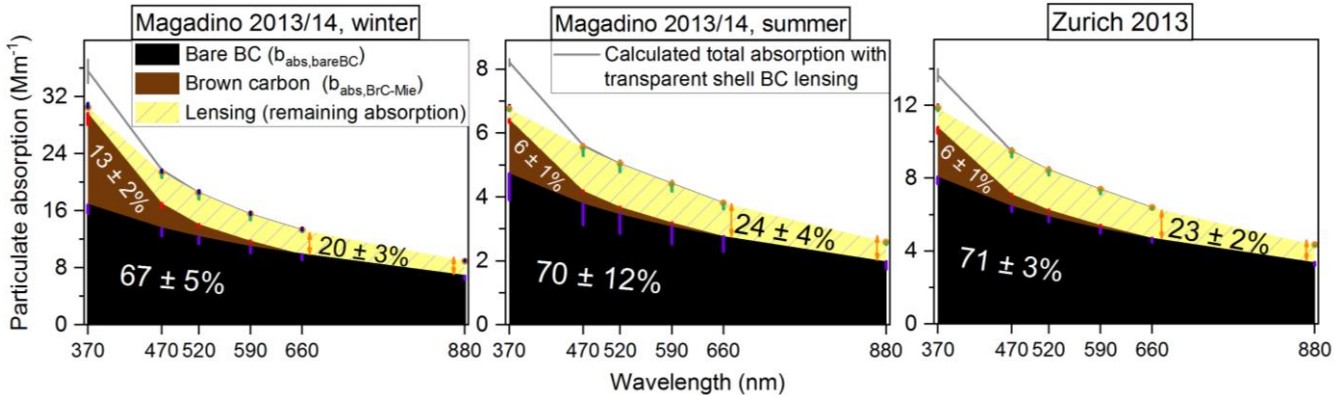

**Figure 6: Optical closure results at six AE33 wavelengths (x-axis, linear scale), shown as cumulative contributions (labels) to total (calibrated) measured particulate absorption (y-axis) of the different carbonaceous aerosol species (bare BC, extractable BrC) and the filter-based BC absorption enhancement (lensing) defined as "remaining absorption". The total measured absorption, $b_{abs,total}$, is indicated with the orange circles (uncertainty: upward blue error bars). Note the different ranges of $b_{abs}$ values (y-axis) between the three cases. Orange arrows indicate the lensing contribution at longer wavelengths (660 nm, 880 nm) for the three cases, with filter-based $E_{abs,BC}$ being equal to (1.35 ± 0.10, 1.33 ± 0.10), (1.37 ± 0.24, 1.38 ± 0.17) and (1.35 ± 0.07, 1.33 ± 0.07) for Magadino winter, Magadino summer and Zurich, respectively. The error bars (minus direction shown; violet: BC; green: BC lensing; red: BrC) were computed by error propagation on the mean values [Eq. (C1)-(C3)] using the $C_\lambda$ (Fig. A1), $MAC_{bareBC,660nm}$ (Fig. S14), $AAE_{bareBC}$ (Fig. 4a) and $b_{abs,BrC-Mie}$ (Text S5) relative errors (Appendix C). The gray line indicates calculated total absorption by adding the absorption by BrC and absorption by BC including transparent shell lensing (assumed to be wavelength-independent) extrapolated from 660 nm.**

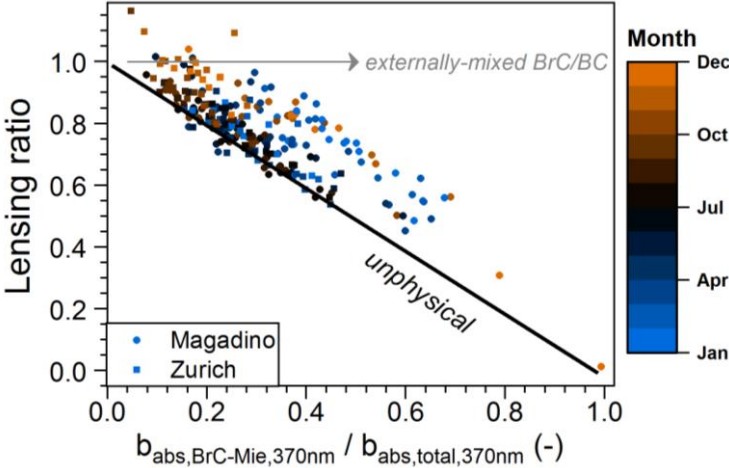

**Figure 7: Relationship between the filter-based lensing ratio, $E_{abs,370nm}/E_{abs,660nm}$ (Eq. (3); independent of MAC$_{bareBC}$), and the estimated average contribution of BrC to total absorption ($b_{abs,BrC-Mie}/b_{abs,total}$) at 370 nm, showing that the extent of filter-based lensing suppression is related to the presence of BrC. Solid black line is the –x+1 line below which data are physically forbidden (where the +1 intercept is based on the assumption that lensing by transparent shells is wavelength-independent). The addition of externally-mixed BrC would result in an increase in the $b_{abs,BrC-Mie}/b_{abs,total}$ without a decrease in the lensing ratio, as indicated by the gray arrow. As a consequence, points near the black line are theoretically impossible if a substantial fraction of BrC is externally mixed from BC.**

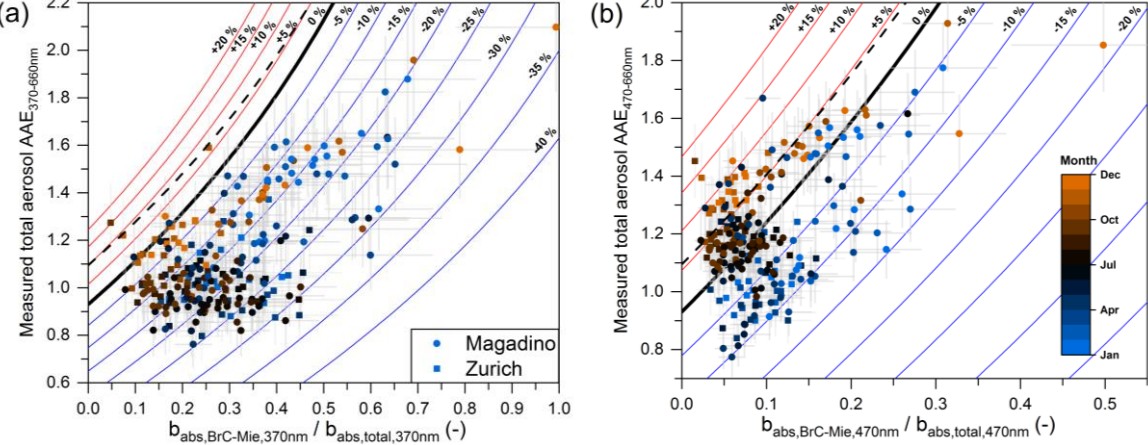

**Figure 8: Total measured aerosol AAE (referenced to 660 nm) vs. the ratio of calculated BrC absorption to measured total absorption at 370 nm (a) and 470 nm (b), showing the effect of BrC/BC interactions on the aerosol absorption profile. Uncertainty in the ratio $C_{370nm}/C_{660nm}$ used for the calibration of AE33 attenuation measurements translates into vertical error bars, while horizontal error bars include uncertainties in AE33 absolute calibration coefficients $C_{370nm}$ and in the BrC absorption at 370 nm from Mie calculations without contingency for errors associated with assuming homogeneous spheres. The isopleths indicate the extent of total absorption under- or over-estimation on the basis of a predicted absorption by BC and BrC, where (apparent) lensing by transparent BC coatings (using AAE$_{bare\ BC}$ = 0.93) is assumed to be wavelength-independent [isopleth = (measured – predicted) / predicted]. The dashed line, which corresponds to additive absorption if transparent shell lensing was increasing with decreasing wavelength (by 10 % or 6 % from 660 nm to 370 nm or 470 nm, respectively), serves to illustrate sensitivity of the closure to spectral extrapolation of transparent shell lensing. Over-predictions (negative isopleths) provide indication for a filter-based lensing suppression effect.**

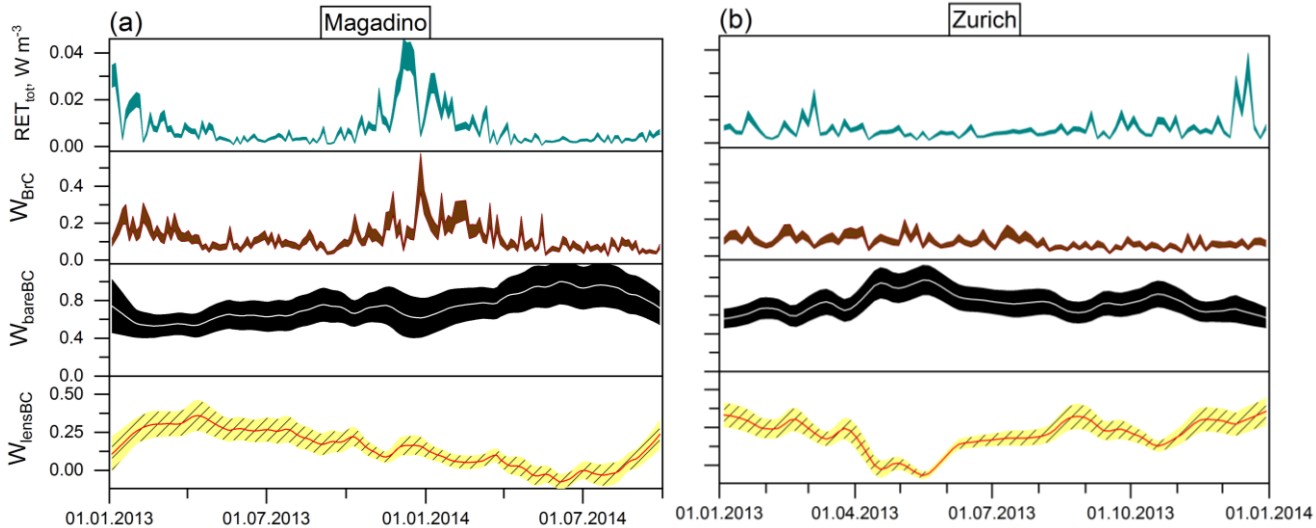

Figure 9: Time series of the extractable Mie-predicted particulate BrC (brown), bare BC (black) and filter-based BC lensing (sparse light yellow) fractional contributions ($W$) to the rate of energy transfer per volume due to light absorption by total carbonaceous aerosol ($RET_{tot}$; dark cyan), upon integration of the respective absorption spectra with the solar radiation spectrum at 300-900 nm (Eq. (D1)), for Magadino (a) and Zurich (b). Contributions are shown as surfaces indicating propagated errors covering the full range upon integration. Contributions by bare BC and lensing were smoothed for demonstration (central estimates are shown as white and red lines, respectively).