# Peer review of "Source-specific light absorption by carbonaceous components in the complex aerosol matrix from yearly filter-based measurements"

_Atmospheric Chemistry and Physics, 2020_

## Referee Comment (RC1) · Anonymous Referee #1 · 12 Jan 2021

This manuscript concerns the optical properties of black carbon (BC) and brown carbon (BrC): significant (if not major) contributor to light-absorption by aerosols. The climate effects of these components are highly uncertain, while the optical properties are an important contributor to these uncertainties. As such, this manuscript addresses a topical subject.

The authors use a number of different, complementary techniques suitable for the purpose, and arrive at important conclusions for the field. However, the presentation of these should be condensed, emphasized and clarified.

The data is a few years old – but this is of little concern here. I like the comparison between an urban and rural location. However, I would like to see a stronger emphasis of site-specifics when comparing to other sites around the world. Further, I think a deepened discussion regarding an underlying assumption of this work: that composite aerosol absorption by different components is additive – comments below.

Below I list a number of major and minor comments. If carefully addressed I think this contribution should be suitable for publication in ACP.

Major comments:

An underlying assumption in this, and also several similar other studies (including citations in the manuscript) is that the absorption by, e.g., different BrC compones, or BrC/BC mixtures is additive. Example: isolated BC abs + isolated BrC abs = abs of mixture.  This might be a fair first order approximation for many applications. Not only considering particle level effects (including scattering effects, e.g., lensing), but also properties at a molecular level. However, a complication with, e.g., BrC, may be illustrated by the almost universally observed power-law like spectrum (as for BC), which suggests that the underlying absorption mechanism is not simply the addition of a large number of different chromophores (which would produce a multi-peak spectrum), but that the absorption is also modulated by interactions of the chromophores with other components; extending to the limit of the graphene-like structure of BC (Andersson, 2017). Empirical evidence shows that this is the case: interactions of BrC chromophores with other components may significantly enhance absorption, e.g., through charge transfer complexes (Phillips and Smith, 2014a,b). Or in other words: the absorption of the mixture is more than the sum of the parts, also at a molecular level. This argument is further, at least to some extent, supported by the observation that absorbing BrC molecules in solutions are much larger (>1000 Da; e.g., increase of molecular size during transport in biomass plumes, associated with loss of absorption by smaller molecules, Di Lorenzo and Young, 2016; Di Lorenzo et al., 2017) than often explored (in laboratory and ambient studies) single-molecule chromophores such as PAHs; nitro-aromatics and similar. Along these lines, the nature of these larger molecular units (large molecules or aggregates?) suggests structural settings by which individual chromophores are likely to interact with surroundings, differently then, e.g., through free translational diffusion. Furthermore, it is not too far-fetched to consider interactions (e.g., charge transfer) between BC and BrC in mixture.

For the purpose of this paper I think the first order approximation of additive absorption is ok. However, I would like to see a discussion about potential limitations of these assumptions - ranging from source apportionment techniques to differentiation of BrC and BC in mixtures - including above references.

Overall the paper is a bit hard to read: many numbers, abbreviations and technical details. I count 135 references, suggesting a thorough literature review, but at the same time indicating a potentially too broad scope for a single paper (more like a review) – with the purpose of clarity and transparence as guiding principles. Perhaps splitting into separate BrC and BC papers would be a possibility. However, this is not a 'reviewer' recommendation, but a thought. In the meantime, I think the authors should revise the text with the goal of a more accessible presentation. Squeeze out the essence (conclusions and interpretations, rather than numerics and technical details) and put the rest into the SI.

Minor comments:

Abstract:

Along with the R&D I find this too long, with too many numbers. I think the reason why, e.g., many high impact journals (e.g., Science or Nature) promotes short abstracts is valid in general: effective and clear communication. I suggest to cut to half its length.

It is not clear to me why the authors focus on the imaginary refractive index as a measure for absorption, when most other studies on BC and BrC report mass absorption cross-section (MAC) or mass absorption efficiency (MAE; different names – same thing). It is clear that the imaginary index goes into the Mie Calculations, but I think it would be much preferred to report MACs etc, especially in Abstract to allow direct comparison with existing literature.

Tar balls appears important in certain regions, while then appears absent here. Is this a key result for the abstract? Further (R&D): what is the significance of this, besides an empirical finding? Does this change the spectral properties in any way? Why are they absent? – sources?

M&M:

Line 50: the absorption of BC is not wavelength-independent. That would suggest constant absorption over all wavelengths, meanwhile it is typically described by a power law, with a variable exponent. Remove.

Line 57: I think the mixing state for BrC (lensing etc) may be equally important for BrC.

Line 63: 'Certain studies suggest that aging leads to bleaching, whereas others suggest formation of secondary BrC'. There is no fundamental reason as to why both cannot happen, and do happen, to different extents, in different environments. I think rephrase to remove possibility of misinterpretation: there is no implicit contradiction in these results.

Line 74: What do you mean by the 'aethalometer model' here? I assume it is not the actual instrument model, but some kind of source apportionment approach. Rephrase.

Line 80: I would expect that it is quite unclear how it responds to photo-chemical aging as well (e.g., the cited Dasari et al., 2019).

Line 84: 'The first consists of a direct estimation of total BrC absorption online by subtraction.' This sentence is unclear: subtraction of what from what? From what data? Etc.

Line 91: What other non-refractory material is intended here?

Line 129: I am not familiar with the MWAA instrument. From the SI appears that this instrument is a filter-based instrument – like the aethalometer. Although probably an improved technology, being filter-based it has the same principal disadvantages regarding measurements of absorption as the aethalometer. Since the MWAA is used for calibration (of the multiple scattering parameter, C) of the aethalometer, I wonder – how is the MWAA calibrated towards non-filter-based methods, e.g., photo-acoustics? Why is this a reliable reference? I am sure this is outlined in the original reference, but it is of importance here and should be clarified.

Section 2.2:

I think this section reads much more technical than what it actually is. Aim to make clearer.

Line 182: The word matrix is mentioned three times in one sentence. But what is this matrix? Do you mean the absorption spectra (dimension 1: wavelength; dimension 2: data point)? Be more concrete.

Line 186: What is meant by: 'This approach provides MAE specific to an OA factor in hypothetical pure form.'?

Line 198: I suppose the MIE calculations are based on size distributions from DMPS? Clarify.

Section 2.3.

I wonder about the fundamental assumption here: that the absorbing species are equally distributed through the particle sizes: this topic was discussed in detail in the cited, Liu et al., 2013. I would like to see a slightly expanded discussion on this compared to the different scenarios explored for different size regimes.

R&D:

Section 3.1.

To provide context, I suppose it would be good to start out with a brief presentation of the study sites, prevailing meteorology (precipitation? Air mass origins? Seasonality? etc), topography or other relevant conditions. I find that – in general - comparisons in the BC/BrC literature tend to aim for general conclusions, while site-specific variability may play a very large role. For instance, comment on line 63.

Further, start out with discussing concentration data, rather than ratios. Even though I am familiar with BrC literature, these ratios mean little to me a priori. This also goes for Figure 1.

Line 232: The means for the winter is indeed lower compared to summer. Meanwhile, the numerical ranges are strongly overlapping. Are these differences statistically significant?

Line 250: PMF provides 'principal components' – which are mathematical deconvolutions. However, these then need to be interpreted in terms of different sources somehow. How was this done? How are HOA, fOOA, SOOA etc attributed to these?

Line 317: The significance of tar balls in this context is unclear.

Lines 355 and forward: Comparisons of MAC are tricky. Different methods and different parametrizations are often used: please discuss these features, and how they relate to the present methodology.

Lines 378-384: I think these important comparisons are glanced over too quickly here. Please see comments on Figure 5 and elaborate.

Summary:

Line 480: If using the word holistic, I would say this study attempts, rather than provides.

The emphasis on biomass influence here, and elsewhere, is not clear to me: do the authors believe this is a central motivation for conducting this study? Or should one interpret this as one of the central findings? The later interpretation precludes the following sentences.

Figure 1.

As a first figure I would like to see the 'raw data' time-series data (absorption coefficients; WSOC concentrations etc), before moving into more convoluted forms, e.g., ratios.

Figure 2.

Overall a good figure, but a bit cluttered. Are you sure all of these data points are essential, or could parts of this figure be presented in the SI?

Figure 4. Try to make the square boxes equal in size.

Figure 5.

I appreciate this comparison, based on geographical variability. Since East and South Asia are quite distinct in terms of the aerosol regime, I would separate this into two groups.

I would also like to see separation w.r.t. atmospheric transport times (e.g., photochemical aging):

For instance, differentiating between near-source (e.g., urban) vs receptor sites; can we pick up a trend?

References to add: Cui et al. (2016); Chen et al. (2017).

Figure 8.

Avoid statements like: 'a new framework'. Just write out what this figure is about.

Appendices:

Line 573: What is the origin of the equation for relative error of EC? What does the number 0.03 signify and how did you arrive at this?

Line 640: What is Zendo? Please provide a link or similar.

References:

Andersson, A (2017) A Model for the Spectral Dependence of Aerosol Sunlight Absorption. ACS Earth Space Chem. DOI: 10.1021/acsearthspacechem.7b00066

Chen et al. (2017) Light absorption enhancement of black carbon from urban haze in Northern China winter. Environ. Poll. DOI: 10.1016/j.envpol.2016.12.004

Cui et al. (2016) Radiative absorption enhancement from coatings on black carbon aerosols. STOTEN. DOI: 10.1016/j.scitotenv.2016.02.026

Di Lorenzo, RA and Young, CJ (2016) Size separation method for absorption characterization in brown carbon: Application to an aged biomass burning sample. GRL. doi:10.1002/2015GL066954.

Di Lorenzo, RA, et al. (2017) Molecular-Size-Separated Brown Carbon Absorption for Biomass-Burning Aerosol at Multiple Field Sites. ES&T. DOI: 10.1021/acs.est.6b06160

Phillips, SA and Smith, GD (2014a) Light Absorption by Charge Transfer Complexes in Brown Carbon Aerosols. ES&T. doi: 10.1021/ez500263j

Phillips, SA and Smith, GD (2014b) Further Evidence for Charge Transfer Complexes in Brown Carbon Aerosols from Excitation−Emission Matrix Fluorescence Spectroscopy. J. Phys. Chem. DOI: 10.1021/jp510709e.

---

## Referee Comment (RC2) · Anonymous Referee #2 · 14 Jan 2021

This manuscript by Moschos et al. carefully described light absorption properties from various sites and sources. They also provide calculations of the optical properties. By combing the experimental and theoretical results, they provide useful conclusions regarding brown carbons and BC lensing effects. These results are useful for further studies regarding radiative forcing and climate. Thus, I recommend this manuscript to be published in ACP. This study provides careful and comprehensive data and figures. All figures are carefully prepared and look nice. On the other hand, the figures include too much information, and I had a difficult time interpreting the meanings. For example, Fig 8 includes Y axis, X axis consisting of two parameters, color and shape of each plot, contour lines, and reference lines (dot line). Although I agree that the figures are useful

and accurate, they are complicated to understand. It is just a suggestion, and the current figures are fine as is but may be improved by simplifying.

Specific comments:

Line 329: The reference by Alexander et al. (2008) is not adequate here because the literature does not show tar-balls but only discussed the possibility of tar-balls.

Line 330: inorganic component. Fig S13 shows a Fe-bearing particle, which may not be a representative of "inorganic" particles.

Line 330: "Pseudospherical" can be a deformed particle that had been liquid in air and deformed on the filter when collected.
* * *

---

## Referee Comment (RC3) · Anonymous Referee #3 · 30 Jan 2021

The manuscript attempts to resolve almost all issues of carbonaceous light absorption based on annual filter-based measurements at two locations. It relies an immense number of references just as a review paper, but in this case the references are actively engaged in the discussion of the methodology applied. That makes the manuscript extremely difficult to follow and understand. Despite its complexity, the manuscript has little if any novelty, it is like a demonstration of all techniques related to filter-based absorption measurements over the past 20 years. Instead of using the more accepted concept of light absorbing carbon continuum, the manuscript relies on the simplified concept of BrC vs BC with spectrally resolvable absorption properties. This is a simple yet quite an established methodology for estimating BB vs FF contributions to PM with

all its inherent biases and uncertainties. Due to the latter, this filter-based approach can by no means yield results that may be used for proving a hypothesis regarding aerosol mixing state and morphology. The statement that the 'first experimental evidence is provided for the suppression of lensing effect by BrC' is simply not supported by the optical closure calculations at shorter wavelength between solvent-based and filter-based optical measurements. Both techniques have a number of limitations and uncertainties exhaustively discussed in the scientific literature, and both are based on several simplifying assumptions which may not necessarily be valid. In the light of these facts, the residual term of optical closure calculations between two fundamentally different measurement methods can by no means signify any 'lensing effect'. The authors themselves devote detailed discussions to (even non-quantifiable) uncertainties and simplifying assumptions (some of them can equally be biases), yet they do not come to the conclusion that a small residual term in closure calculations is well within the range of uncertainties and should not be over-interpreted. Optical closure calculations are based on the assumption that total filter absorption, absorptions in methanol and water extract, and residual absorption on filter (after solvent extraction) are all additive. Is it possible that solvent extraction changes some properties on filter that affect absorption measurements (e.g. scattering effect)? Is there any hysteresis of solvent extraction (e.g. residual water that affects measurements)? Additivity means that if a spot of a loaded filter is measured for absorption, then extracted in water and methanol, then the extracts are uniformly redispersed on the residual filter spot and the solvents are evaporated, and the filter spot is again measured for absorption, we get exactly the same absorption as for the untreated filter. I never did such a simple experiment but I would seriously doubt that the two values would be identical.

---

## Author Comment (AC1) · 6 May 2021

**Responses to the comments of Referee #1**

We thank Referee #1 for the valuable comments, which have greatly helped us to improve the manuscript. Please find below our point-by-point responses (in blue normal font) to the comments of Referee #1 (*in black italic*). The changes in the revised manuscript are in green.

*This manuscript concerns the optical properties of black carbon (BC) and brown carbon (BrC): significant (if not major) contributor to light-absorption by aerosols. The climate effects of these components are highly uncertain, while the optical properties are an important contributor to these uncertainties. As such, this manuscript addresses a topical subject.*

*The authors use a number of different, complementary techniques suitable for the purpose, and arrive at important conclusions for the field. However, the presentation of these should be condensed, emphasized and clarified.*

*The data is a few years old – but this is of little concern here. I like the comparison between an urban and rural location. However, I would like to see a stronger emphasis of site-specifics when comparing to other sites around the world. Further, I think a deepened discussion regarding an underlying assumption of this work: that composite aerosol absorption by different components is additive – comments below.*

*Below I list a number of major and minor comments. If carefully addressed I think this contribution should be suitable for publication in ACP.*

We thank the referee for the constructive comments, which we have addressed below.

With regard to the comment that the data are a few years old, these sites were chosen because of the available detailed chemical and source apportionment information. We have analyzed two sites that cover the major and most common aerosol sources found in Europe: primary biomass burning, secondary anthropogenic and biogenic organic aerosols, traffic emissions, and secondary inorganic components. Please find our replies to the specific comments below.

**Major comments:**

*An underlying assumption in this, and also several similar other studies (including citations in the manuscript) is that the absorption by, e.g., different BrC components, or BrC/BC mixtures is additive. Example: isolated BC abs + isolated BrC abs = abs of mixture. This might be a fair first order approximation for many applications. Not only considering particle level effects (including scattering effects, e.g., lensing), but also properties at a molecular level. However, a complication with, e.g., BrC, may be illustrated by the almost universally observed power-law like spectrum (as for BC), which suggests that the underlying absorption mechanism is not simply the addition of a large number of different chromophores (which would produce a multi-peak spectrum), but that the absorption is also modulated by interactions of the chromophores with other components; extending to the limit of the graphene-like structure of BC (Andersson, 2017). Empirical evidence shows that this is the case: interactions of BrC chromophores with other components may significantly enhance absorption, e.g., through charge transfer complexes (Phillips and Smith, 2014a,b). Or in other words: the absorption of the mixture is more than the sum of the parts, also at a molecular level. This argument is further, at least to some extent, supported by the observation that absorbing BrC molecules in solutions are much larger (>1000 Da; e.g., increase of molecular size during transport in biomass plumes, associated with loss of absorption by smaller molecules, Di Lorenzo and Young, 2016; Di Lorenzo et al., 2017) than often explored (in laboratory and ambient studies) single-molecule chromophores such as PAHs; nitro-aromatics and similar. Along these lines, the nature of these larger molecular units (large molecules or aggregates?) suggests structural settings by which individual chromophores are likely to interact with surroundings, differently then, e.g., through free translational diffusion. Furthermore, it is not too far-fetched to consider interactions (e.g., charge transfer) between BC and BrC in mixture.*

*For the purpose of this paper I think the first order approximation of additive absorption is ok. However, I would like to see a discussion about potential limitations of these assumptions -ranging from source apportionment techniques to differentiation of BrC and BC in mixtures - including above references.*

In this comment, the reviewer refers to the effect of chemical interactions on the absorption of brown carbon and how this potentially affects the comparison between absorption measurements in diluted solutions and actual absorption in concentrated bulk aerosols. We agree with the reviewer that such effects may affect our results, and provide below a comprehensive argumentation regarding the level of confidence for the occurrence of such effects in our samples considering also previous relevant studies.

The importance of charge transfer complexes on the optical absorption properties of dissolved organic matter is still debated in literature, with some studies showing an important effect (e.g., Phillips and Smith, 2014) and others not (McKay et al., 2017; Trofimova et al., 2019). In our work, we have extensively assessed the solvent effect on attenuation measurements. First, we have extracted the filter samples in multiple solvents covering an extremely wide range of polarities (Fig. S3), but did not observe a clear effect on the OA attenuation. If the charge transfer between BrC molecules and the solvent played a considerable role for their absorption, the change in solvent polarity would alter the attenuation measurement; however, this was not observed. Second, we compared the absorbance of samples extracted in water and then diluted in methanol or in water in a 10:90 ratio, in order to decouple between the solvent effect on OA attenuation and solubility (Sect. 2.1.4). We have not identified a substantial water/methanol solvent effect on the attenuation, consistent with previous literature assessments (Trofimova et al., 2019; Mo et al. 2017; Chen and Bond 2010). Third, we show that the absorbance of the extracts scaled linearly with concentration over a wide range. The lack of/little solvent and concentration effects are an indication that the interactions of the brown carbon molecules with their matrix have little effect on their absorbance. Nevertheless, we cannot rule out potential chemical interactions between airborne BrC molecules in their concentrated media or with BC. This has received far less attention in literature and may contribute to unquantifiable experimental uncertainties relating to the delineation of BC and BrC contributions to mixtures (Andersson, 2017). We have now discussed these effects in Sect. 2.1.4 and Appendix C (non-quantifiable uncertainties).

The following has been added in Sect. 2.1.4:

"The results from (i) the comparison between different solvents, (ii) the dilution series of the MeOH extracts at various concentrations and (iii) the assessment of the water/MeOH solvent effect, indicate that the interactions of the BrC molecules with their matrix have little effect on their absorbance. This is in line with recent findings (McKay et al., 2017; Trofimova et al., 2019)."

The following has been added in Appendix C:

"- Uncertainties related to potential chemical interactions between airborne BrC molecules in their concentrated media or with BC, and their effect on the BC/BrC contributions to the mixture absorption (Andersson, 2017)."

*Overall the paper is a bit hard to read: many numbers, abbreviations and technical details. I count 135 references, suggesting a thorough literature review, but at the same time indicating a potentially too broad scope for a single paper (more like a review) – with the purpose of clarity and transparence as guiding principles. Perhaps splitting into separate BrC and BC papers would be a possibility. However, this is not a 'reviewer' recommendation, but a thought. In the meantime, I think the authors should revise the text with the goal of a more accessible presentation. Squeeze out the essence (conclusions and interpretations, rather than numerics and technical details) and put the rest into the SI.*

We acknowledge that the paper is lengthy. We have first considered splitting the paper in half. However, we felt that all pieces of information are needed to assess the effect of different aerosol components on the aerosol absolute absorption and absorption profile. We consider it necessary to present all the experimental results together with a description of the associated methodological steps to arrive to the conclusions presented in our study. We believe that the paper follows a logical thread and is balanced between the different inter-linked topics addressed. Each subsection has a title that helps the reader to quickly assess whether there is any take-home message in the content to spend time reading it. We

make use of the Appendix for technical details, abbreviations and symbol/equation definitions. Therefore, we would like to keep the manuscript overall in its current form, but at the same time we have simplified the abstract, text and figures whenever possible based on the reviewer comments (see below). We have also reduced the number of the preprint references from 135 to 99 (by 27 %; see list of removed citations in the end of this response).

**Minor comments:**

*Abstract:*
*Along with the R&D I find this too long, with too many numbers. I think the reason why, e.g., many high impact journals (e.g., Science or Nature) promotes short abstracts is valid in general: effective and clear communication. I suggest to cut to half its length.*

The paper contains multiple R&D sub-sections presenting results of equal value for the interpretations and conclusions. The abstract presents concisely these results together with the final conclusions. While we have retained all the elements of the abstract, we have followed the reviewers' suggestion and simplified the result presentation in the abstract, especially those related to the lensing suppression. The new abstract reads as follows:

Understanding the sources of light-absorbing organic (brown) carbon (BrC) and its interaction with black carbon (BC) and other non-refractory particulate matter (NR-PM) fractions is important for reducing uncertainties in the aerosol direct radiative forcing. In this study, we combine multiple filter-based techniques to achieve long-term, spectrally-resolved, source- and species-specific atmospheric absorption closure. We determine the mass absorption efficiency (MAE) in dilute bulk solutions at 370 nm to be equal to 1.4 $m^2\,g^{-1}$ for fresh biomass smoke, 0.7 $m^2\,g^{-1}$ for winter-oxygenated OA, and 0.13 $m^2\,g^{-1}$ for other less absorbing OA. We apply Mie calculations to estimate the contributions of these fractions to total aerosol absorption. While enhanced absorption in the near-UV has been traditionally attributed to primary biomass smoke, here we show that anthropogenic oxygenated OA may be equally important for BrC absorption during winter, especially at an urban background site. We demonstrate that insoluble tar-balls are negligible in residential biomass burning atmospheric samples of this study, and thus could attribute the totality of the NR-PM absorption at shorter wavelengths to methanol-extractable BrC. As for BC, we show that the mass absorption cross-section (MAC) of this fraction is independent of its source, while we observe evidence for a lensing effect associated with the presence of NR-PM components. We find that bare BC has a MAC of 6.3 $m^2\,g^{-1}$ at 660 nm and an absorption Ångström exponent (AAE) of $0.93 \pm 0.16$, while in the presence of coatings its absorption is enhanced by a factor of ~1.4. Based on Mie-calculations of closure between observed and predicted total light absorption, we provide an indication for a suppression of the lensing effect by BrC. The total absorption reduction remains modest, ~10-20 % at 370 nm, and is restricted to shorter wavelengths where BrC absorption is significant. Overall, our results allow an assessment of the relative importance of the different aerosol fractions to the total absorption, for aerosols from a wide range of sources and atmospheric ages. When integrated with the solar spectrum at 300-900 nm, bare BC is found to contribute around two thirds of the solar radiation absorption by total carbonaceous aerosols, amplified by the lensing effect (with an interquartile range, IQR, of 8-27 %), while the IQR of the contributions by particulate BrC is 6-13 % (13-20 % at the rural site during winter). Future studies that will directly benefit from these results include: (a) Optical modelling aiming at understanding the absorption profiles of a complex aerosol composed of BrC, BC and lensing-inducing coatings; (b) Source apportionment aiming at understanding the sources of BC and BrC from the aerosol absorption profiles; (c) Global modelling aiming at quantifying the most important aerosol absorbers.

*It is not clear to me why the authors focus on the imaginary refractive index as a measure for absorption, when most other studies on BC and BrC report mass absorption cross-section (MAC) or mass absorption efficiency (MAE; different names – same thing). It is clear that the imaginary index goes into the Mie Calculations, but I think it would be much preferred to report MACs etc, especially in Abstract to allow direct comparison with existing literature.*

We agree with the reviewer that the MAC or MAE are both popular parameters to quantify absorption. However, the imaginary part of the refractive index is also popular (e.g. Lu et al., 2015) and is closest to the physical quantity which we measured in our UV-Vis samples. Moreover, the prediction of MAC

from *k* requires the assumption of particle size. Nevertheless, in our submitted Fig. S10 we did provide MAEs (assuming Rayleigh regime optics) for our data. In our submitted abstract we did report MAEs at 370 nm.

*Tar balls appears important in certain regions, while then appears absent here. Is this a key result for the abstract? Further (R&D): what is the significance of this, besides an empirical finding? Does this change the spectral properties in any way? Why are they absent? – sources?*

We agree that the significance and relevance of this finding should be more clearly presented. We have modified the Abstract and Sect. 3.4, to provide a clearer picture of the implications for this study, focusing on the fact that insoluble tar carbon could not be identified in this work:

The following has been modified in the Abstract:

"We demonstrate that insoluble tar-balls are negligible in residential biomass burning atmospheric samples of this study, and thus could attribute the totality of the NR-PM absorption at shorter wavelengths to methanol-extractable BrC."

The following have been modified in Sect. 3.4:

"We have conducted (sequential) extractions and MWAA measurements to determine the AAE of uncoated, pure BC and to examine the potential presence of (insoluble) tar carbon. The latter has been observed in atmospheric samples rich in fresh biomass burning/wildfire smoke (China et al., 2013), and exhibits distinct absorption properties than those of BC and conventional BrC (Corbin et al., 2019).
"The absence of tar-balls in our study may be related to the specific source of biomass-burning organics in our region, i.e., wood burning for residential heating in high-efficiency stoves rather than in low-efficiency wildfires, although more measurements of direct emissions would be needed to confirm this."

*M&M:*
*Line 50: the absorption of BC is not wavelength-independent. That would suggest constant absorption over all wavelengths, meanwhile it is typically described by a power law, with a variable exponent. Remove.*

We thank the reviewer for pointing out this language error. We intended to refer to the refractive index of BC. A wavelength-independent refractive index results in a power law absorption with an exponent close to 1.0, which is the case for externally-mixed mature BC. Variation in this exponent is typically minor or due to changes in maturity or BrC content. The relevant sentence now reads as follows, for clarity:

Black carbon (BC) is a strong broadband absorber with a nearly wavelength-independent refractive index and negligible solubility in common solvents (Bond and Bergstrom, 2006; Moosmüller et al., 2009; Petzold et al., 2013; Samset et al., 2018b; Corbin et al., 2019).

*Line 57: I think the mixing state for BrC (lensing etc) may be equally important for BrC.*

We have rephrased the sentence according to the reviewer's comment:

Accounting for the optical absorption properties of BrC, its interaction with BC, and their mixing state with other particulate species are important for reducing the large uncertainty in the aerosol direct radiative forcing (Laskin et al., 2015; Gustafsson and Ramanathan, 2016; Samset et al., 2018b; Saleh, 2020).

*Line 63: 'Certain studies suggest that aging leads to bleaching, whereas others suggest formation of secondary BrC'. There is no fundamental reason as to why both cannot happen, and do happen, to different extents, in different environments. I think rephrase to remove possibility of misinterpretation: there is no implicit contradiction in these results.*

We agree that this sentence could be misinterpreted. We have rephrased the sentence according to the reviewer's comment:

A growing body of evidence suggests that aerosol aging may be associated with both bleaching and/or formation of secondary BrC (Saleh et al., 2013; Zhong and Jang, 2014; Zhao et al., 2014; Kumar et al., 2018; Dasari et al., 2019). However, the net effect of these processes on the atmospheric BrC absorption remains elusive.

*Line 74: What do you mean by the 'aethalometer model' here? I assume it is not the actual instrument model, but some kind of source apportionment approach. Rephrase.*

The reviewer is correct, "Aethalometer model" is not the actual instrument model, but the name of the approach used to apportion equivalent black carbon (eBC) to sources. We have rephrased the sentence for clarity:

An Aethalometer-based source apportionment (SA) model (Sandradewi et al., 2008), hereafter denoted "the Aethalometer SA model", has been proposed to separate the contributions of wood burning and traffic emissions to equivalent BC, eBC (Petzold et al., 2013). The Aethalometer SA model is based on differences in the spectral profiles between the two aerosol sources, with biomass burning emissions characterized by enhanced absorption at shorter wavelengths, or high absorption Ångström exponents (AAE).

*Line 80: I would expect that it is quite unclear how it responds to photo-chemical aging as well (e.g., the cited Dasari et al., 2019).*

We have rephrased the sentence according to the reviewer's suggestion:

However, it is unclear how this model responds to photochemical aging, SOA formation and lensing (Martinsson et al., 2015; Garg et al., 2016; Dasari et al., 2019).

*Line 84: 'The first consists of a direct estimation of total BrC absorption online by subtraction.' This sentence is unclear: subtraction of what from what? From what data? Etc.*

We have rephrased the sentence for clarity:

Two main approaches exist to estimate the absorption by individual aerosol components in heterogeneous atmospheric particle ensembles (Moffet et al., 2010). The first approach is based on online measurements and consists of a direct estimation of BrC absorption at shorter wavelengths by subtracting the estimated total BC absorption (assuming an AAE for bulk BC) from the total aerosol absorption. This rather convenient approach might lead to biased estimates (see Sect. 4), if the decoupling from potentially variable optical properties of pure BC and its absorption enhancement due to coating acquisition is not ensured.

*Line 91: What other non-refractory material is intended here?*

We have complemented this sentence:

With aging, BC from both sources may be further coated with non-refractory particulate matter species (BrC, non-absorbing OA, inorganics).

*Line 129: I am not familiar with the MWAA instrument. From the SI appears that this instrument is a filter-based instrument – like the aethalometer. Although probably an improved technology, being filter-based it has the same principal disadvantages regarding measurements of absorption as the aethalometer. Since the MWAA is used for calibration (of the multiple scattering parameter, C) of the*

*aethalometer, I wonder – how is the MWAA calibrated towards non-filter-based methods, e.g., photo-acoustics? Why is this a reliable reference? I am sure this is outlined in the original reference, but it is of importance here and should be clarified.*

The reviewer is correct, both MWAA and AE33 are filter-based approaches. However, while AE33 only measures light transmission, the MWAA measures the transmission and backward scattering at two fixed angles, which takes into account the attenuation due to the multiple scattering by the filter material. The MWAA is therefore comparable to the MAAP (e.g., see Valentini et al., 2021; Bernardoni et al., 2021; Saturno et al., 2017), and not the AE33, but at five different wavelengths instead of one for the MAAP. The comparison between MWAA and AE33 measurements allows calibrating for the multiple scattering parameter, *C*. We have modified Sect. 2.1.3 by adding the following:

For a selected set of 27 offline samples, we used the multi-wavelength absorption analyzer (MWAA; Massabo et al., 2013) to determine the total aerosol absorption at five wavelengths. The MWAA measures the transmission and backward scattering at two fixed angles, which takes into account the scattering contributions to attenuation. The filter absorbance measured by MWAA has been successfully validated against both a polar photometer and a multi-angle absorption photometer/MAAP (Massabo et al., 2013) which, in turn, has been validated against numerous *in situ* methods.

*Section 2.2:*
*I think this section reads much more technical than what it actually is. Aim to make clearer.*

Please see replies to next comments.

*Line 182: The word matrix is mentioned three times in one sentence. But what is this matrix? Do you mean the absorption spectra (dimension 1: wavelength; dimension 2: data point)? Be more concrete.*

We have modified the sentence as follows:

Briefly, the model minimizes the residual difference between the observed $A_j(\lambda)$ (model input, in Mm$^{-1}$) and a reconstructed $A_j(\lambda)$. The latter is the product of the mass concentration time series ($M$) of each AMS/PMF OA factor (model constraints; in µg m$^{-3}$) and a matrix containing the factor-specific absorption efficiency spectra, MAE($\lambda$) (model output; in m$^2$ g$^{-1}$).

*Line 186: What is meant by: 'This approach provides MAE specific to an OA factor in hypothetical pure form.'?*

We have modified the sentence as follows:

This approach provides MAE specific to an OA factor in hypothetical pure form, i.e., extracted in a solvent and externally mixed from other aerosol components. This approach may facilitate optical (e.g., radiative forcing) calculations starting from fundamental, intensive material properties.

*Line 198: I suppose the MIE calculations are based on size distributions from DMPS? Clarify.*

We have extended the discussion to clarify that SMPS measurements were considered for optical calculations. The overall passage now reads as follows:

We used Mie calculations (Bohren and Huffman, 1998) to estimate the BrC absorption, $b_{abs,BrC-Mie,j}(\lambda)$, at four AE33 wavelengths (370 nm, 470 nm, 520 nm, and 590 nm) with Mie code programmed in the software package Igor Pro (WaveMetrics). The main inputs required for Mie calculations of light absorption coefficients are the particle size distribution (partially constrained here from SMPS measurements; Sect. 2.1.2) and the refractive index of the aerosol material in question. […]

*Section 2.3.*

*I wonder about the fundamental assumption here: that the absorbing species are equally distributed through the particle sizes: this topic was discussed in detail in the cited, Liu et al., 2013. I would like to see a slightly expanded discussion on this compared to the different scenarios explored for different size regimes.*

We do not have a handle on the distribution of BrC matter on the different particles. Therefore, we have tested two extreme cases of "small particles" (~120 nm) and "large particles" (200-400 nm) in our BrC modelling, as detailed in Sect 2.3 and Text S5.

*R&D:*
*Section 3.1.*
*To provide context, I suppose it would be good to start out with a brief presentation of the study sites, prevailing meteorology (precipitation? Air mass origins? Seasonality? etc), topography or other relevant conditions. I find that – in general - comparisons in the BC/BrC literature tend to aim for general conclusions, while site-specific variability may play a very large role. For instance, comment on line 63.*

Based on the reviewer comment, we have added the following sentence in Sect. 2.1.1 in Materials and Methods, in addition to information already provided in Text S1:

Magadino is a rural background site affected in winter by intense wood burning activity for residential heating, whereas Zurich is an urban background site affected by regional transport of anthropogenic-dominated pollution in winter and spring. Both sites are affected by biogenic secondary emissions in summer (see Text S1; Fig. S2).

*Further, start out with discussing concentration data, rather than ratios. Even though I am familiar with BrC literature, these ratios mean little to me a priori. This also goes for Figure 1.*

See the reply below on the comment about Fig. 1.

*Line 232: The means for the winter is indeed lower compared to summer. Meanwhile, the numerical ranges are strongly overlapping. Are these differences statistically significant?*

We revised the text throughout and now focus on commenting on systematic differences in the ratios (i.e., seasonal averages), of which the standard error is significantly lower (due to the large number of samples) compared to the day-to-day variability.

*Line 250: PMF provides 'principal components' – which are mathematical deconvolutions. However, these then need to be interpreted in terms of different sources somehow. How was this done? How are HOA, fOOA, SOOA etc attributed to these?*

We have added the following sentence in the beginning of this sub-section for clarification:

We have applied UV/Vis-PMF (Sect. 2.2) to infer the absorption properties of methanol-soluble OA factors previously identified from offline AMS/PMF analyses (Fig. S2; Daellenbach et al., 2017; Vlachou et al., 2018).

*Line 317: The significance of tar balls in this context is unclear.*

We have modified the associated text to provide a clearer context for referring to tar BrC, with regard to the wavelength-dependence of its absorption:

The resulting AAE is consistent with that of pure BC across all wavelengths, indicating that the absorption at shorter wavelengths is not dominated by insoluble biomass burning tar carbon in this study, but by MeOH-soluble BrC. Previous laboratory work using the same technique observed that

insoluble tar-balls, with AAE values comparable to those of soluble BrC, can dominate the BrC absorption from residual fuel combustion in marine ship engines (Corbin et al., 2019).

*Lines 355 and forward: Comparisons of MAC are tricky. Different methods and different parametrizations are often used: please discuss these features, and how they relate to the present methodology.*

We have discussed the features suggested by the referee, as follows:

The value of $MAC_{bareBC,880nm}$ is consistent with that reported in a laboratory study of externally-mixed BrC/BC emissions from residential wood-burning (4.7 ± 0.3 $m^2$ $g^{-1}$; Kumar et al., 2018), calculated as the slope of a linear fit between MWAA-calibrated Aethalometer attenuation values vs. the Sunset-EC mass. Also, a review of ten recent direct measurements of absorption and mass with different in-situ instruments (Liu et al., 2020) concluded that uncoated (freshly-emitted) BC has a typical MAC of 6.6 ± 0.6 $m^2$ $g^{-1}$ at 660 nm (extrapolated from 8.0 ± 0.7 $m^2$ $g^{-1}$ at 550 nm using $AAE_{bareBC}$ = 1.0), which is within one standard deviation of the value recommended earlier (Bond and Bergstrom, 2006).

*Lines 378-384: I think these important comparisons are glanced over too quickly here. Please see comments on Figure 5 and elaborate.*

We have provided a reply below with the comment on Fig. 5.

*Summary:*
*Line 480: If using the word holistic, I would say this study attempts, rather than provides.*

We have rephrased the sentence as follows:

"This study attempted to provide a holistic approach to understand…"

*The emphasis on biomass influence here, and elsewhere, is not clear to me: do the authors believe this is a central motivation for conducting this study? Or should one interpret this as one of the central findings? The later interpretation precludes the following sentences.*

We agree with the referee that the biomass influence is not a central motivation of this study, so the emphasis has been removed from the conclusions. The modified text reads as follows:

This study attempted to provide a holistic approach to understand the spectrally-resolved absorption by atmospheric BrC and BC, using long time series of daily samples from filter-based measurements.

*Figure 1.*
*As a first figure I would like to see the 'raw data' time-series data (absorption coefficients; WSOC concentrations etc), before moving into more convoluted forms, e.g., ratios.*

The raw data for WSOA had been shown in Moschos et al. (2018), while the total OA factor mass time series were provided in Daellenbach et al. (2017) (also in Fig. S2 of this manuscript). Therefore, we have chosen not to show the raw data once again in the main figures of this manuscript. However, we have followed the reviewer recommendation and now discuss in the text raw mass data before discussing ratios in Fig. 1. We also consider the Mie-predicted BrC absorption coefficients, discussed in Sect. 3.3 and presented Fig. 3, more pertinent for this study than solution-phase absorbance raw data. The following was added in Sect. 3.1:

The WSOA (total OA) average concentration is 3.7 (5.4) and 6.0 (9.4) µg $m^{-3}$ in summer and winter respectively, accounting for ~66 % of the total OA.

*Figure 2.*
*Overall a good figure, but a bit cluttered. Are you sure all of these data points are essential, or could parts of this figure be presented in the SI?*

We have updated the figure upon removal of certain literature data points, by retaining the balance between laboratory and field studies and sufficient number of studies that are comparable to each UV/Vis-PMF spectrum.

[Figure]

Figure 2: Spectrally-resolved median $k$ values (MAE in Fig. S10) of the different methanol-soluble (water-soluble: Moschos et al., 2018) offline AMS/PMF-based OA source components [(WINS-)BBOA corrected for extraction efficiency in MeOH; WINS-BBOA corrected for the water/methanol solvent effect; Other OA: HOA+COA+SOOA+fOOA], together with the IQR from different UV/Vis-PMF sensitivity runs (Text S4 & Table S2). The retrieved $k_{BrC}$ are compared to those of previous laboratory and field studies focusing on methanol-extractable or total organics [a: Chen & Bond (2010), b: Xie et al. (2017), c: Li et al. (2019), d: Liu et al. (2015), e: Lu et al. (2015), f: Adler et al. (2019), g: Corbin et al. (2019), h: Kirchstetter et al. (2004), i: Lack et al. (2012), j: Cappa et al. (2019), k: Yan et al. (2020), l: Cappa et al. (2012), m: Moschos et al. (2018)].

*Figure 4.*
*Try to make the square boxes equal in size.*

We have updated the figure using the same size for the panel boxes.

*Figure 5.*
*I appreciate this comparison, based on geographical variability. Since East and South Asia are quite distinct in terms of the aerosol regime, I would separate this into two groups.*
*I would also like to see separation w.r.t. atmospheric transport times (e.g., photochemical aging):*
*For instance, differentiating between near-source (e.g., urban) vs receptor sites; can we pick up a trend?*
*References to add: Cui et al. (2016); Chen et al. (2017).*

We have separated into the two suggested groups: South Asia and East Asia, including the two references suggested by the referee (see updated figure below), and added the following sentence:

The average $E_{abs,BC}$ at near-source/urban sites is slightly lower than the global average; there is no increasing trend towards remote sites far from any direct emission influence, possibly due to measurement errors, calibration uncertainties and/or the collapse of the BC "core" for high proxy values.

[Figure]

Figure 5: Summary of campaign-average (error bars: ± 1 SD) filter-based BC absorption enhancement factor (Eq. (C4)) at longer wavelengths (> 600 nm) reported or inferred, using/extrapolating an observational reference value for uncoated BC MAC (Bond and Bergstrom, 2006), using the most common instruments (Aethalometer, PSAP, MAAP, Sunset/DRI, OCEC analyzer) at various locations/years. The gray-shaded area shows the global average within 1 SD (1.5 ± 0.3). The red horizontal dashed line denotes the full dataset average $E_{abs,BC}$ obtained in this study > 600 nm (1.36) where BrC does not contribute to absorption (Fig. 1-3).

*Figure 8.*
*Avoid statements like: 'a new framework'. Just write out what this figure is about.*

The first sentence of Fig. 8 caption now reads:

"Total measured aerosol AAE (referenced to 660 nm) vs. the ratio of calculated BrC absorption to measured total absorption at 370 nm (a) and 470 nm (b), showing the effect of BrC/BC interactions on the aerosol absorption profile."

*Appendices:*
*Line 573: What is the origin of the equation for relative error of EC? What does the number 0.03 signify and how did you arrive at this?*

The error on the EC mass has been defined as: $\sqrt{(0.03 \times OC)^2 + (0.03 \times EC)^2}$. The number 0.03 signifies a 3 % variability in these quantities. Virtually all EC measurements were above the noise level. The equation has been determined empirically based on comparisons of repeated measurements in our lab. It takes into account the errors related to the quantification of EC and the separation between OC and EC.

*Line 640: What is Zendo? Please provide a link or similar.*

Zenodo (https://zenodo.org) is a general-purpose open-access repository developed under the European OpenAIRE program and operated by CERN. It allows researchers to deposit research papers, data sets, research software, reports, and any other research related digital artifacts. For each submission, a persistent digital object identifier (DOI) is minted, which makes the stored items easily citeable.

*References:*
*Andersson, A (2017) A Model for the Spectral Dependence of Aerosol Sunlight Absorption. ACS Earth Space Chem. DOI: 10.1021/acsearthspacechem.7b00066*

*Chen et al. (2017) Light absorption enhancement of black carbon from urban haze in Northern China winter. Environ. Poll. DOI: 10.1016/j.envpol.2016.12.004*

*Cui et al. (2016) Radiative absorption enhancement from coatings on black carbon aerosols. STOTEN. DOI: 10.1016/j.scitotenv.2016.02.026*

*Di Lorenzo, RA and Young, CJ (2016) Size separation method for absorption characterization in brown carbon: Application to an aged biomass burning sample. GRL. doi:10.1002/2015GL066954.*

*Di Lorenzo, RA, et al. (2017) Molecular-Size-Separated Brown Carbon Absorption for Biomass-Burning Aerosol at Multiple Field Sites. ES&T. DOI: 10.1021/acs.est.6b06160*

*Phillips, SA and Smith, GD (2014a) Light Absorption by Charge Transfer Complexes in Brown Carbon Aerosols. ES&T. doi: 10.1021/ez500263j*

*Phillips, SA and Smith, GD (2014b) Further Evidence for Charge Transfer Complexes in Brown Carbon Aerosols from Excitation−Emission Matrix Fluorescence Spectroscopy. J. Phys. Chem. DOI: 10.1021/jp510709e.*

**References** (not included in the preprint/revised manuscript):

Bernardoni, V., Ferrero, L., Bolzacchini, E., Forello, A. C., Gregorič, A., Massabò, D., Močnik, G., Prati, P., Rigler, M., Santagostini, L., Soldan, F., Valentini, S., Valli, G., and Vecchi, R.: Determination of aethalometer multiple-scattering enhancement parameters and impact on source apportionment during the winter 2017/18 EMEP/ACTRIS/COLOSSAL campaign in Milan, Atmos. Meas. Tech., 14, 2919-2940, 10.5194/amt-14-2919-2021, 2021.

Phillips, S. M., and Smith, G. D.: Light absorption by charge transfer complexes in brown carbon aerosols, Environ. Sci. Technol. Lett., 1, 382-386, 10.1021/ez500263j, 2014.

Saturno, J., Pöhlker, C., Massabò, D., Brito, J., Carbone, S., Cheng, Y., Chi, X., Ditas, F., Hrabě de Angelis, I., Morán-Zuloaga, D., Pöhlker, M. L., Rizzo, L. V., Walter, D., Wang, Q., Artaxo, P., Prati, P., and Andreae, M. O.: Comparison of different aethalometer correction schemes and a reference multi-wavelength absorption technique for ambient aerosol data, Atmos. Meas. Tech., 10, 2837-2850, 10.5194/amt-10-2837-2017, 2017.

Valentini, S., Bernardoni, V., Bolzacchini, E., Ciniglia, D., Ferrero, L., Forello, A. C., Massabó, D., Pandolfi, M., Prati, P., Soldan, F., Valli, G., Yus-Díez, J., Alastuey, A., and Vecchi, R.: Applicability of benchtop multi-wavelength polar photometers to off-line measurements of the multi-angle absorption photometer (MAAP) samples, J. Aerosol Sci., 152, 10.1016/j.jaerosci.2020.105701, 2021.

**References that were removed**, according to the reviewer's comment:

Alexander, D. T. L., Crozier, P. A., and Anderson, J. R.: Brown Carbon Spheres in East Asian Outflow and their Optical Properties, Science, 321, 833–836, 2008.

Barnard, J. C., Volkamer, R., and Kassianov, E. I.: Estimation of the mass absorption cross section of the organic carbon component of aerosols in the Mexico City metropolitan area, Atmos. Chem. Phys., 8, 6665-6679, 10.5194/acp-8-6665-2008, 2008.

Bernardoni, V., Pileci, R., Caponi, L., and Massabò, D.: The multi-wavelength absorption analyzer (MWAA) model as a tool for source and component apportionment based on aerosol absorption properties: Application to samples collected in different environments, Atmosphere, 8, 10.3390/atmos8110218, 2017.

Chen, L. W. A., Chow, J. C., Wang, X. L., Robles, J. A., Sumlin, B. J., Lowenthal, D. H., Zimmermann, R., and Watson, J. G.: Multi-wavelength optical measurement to enhance thermal/optical analysis for carbonaceous aerosol, Atmos. Meas. Tech., 8, 451-461, 10.5194/amt-8-451-2015, 2015.

Cheng, Y., He, K. B., Zheng, M., Duan, F. K., Du, Z. Y., Ma, Y. L., Tan, J. H., Yang, F. M., Liu, J. M., Zhang, X. L., Weber, R. J., Bergin, M. H., and Russell, A. G.: Mass absorption efficiency of elemental carbon and water-soluble organic carbon in Beijing, China, Atmos. Chem. Phys., 11, 11497-11510, 10.5194/acp-11-11497-2011, 2011.

Coz, E., and Leck, C.: Morphology and state of mixture of atmospheric soot aggregates during the winter season over Southern Asia-a quantitative approach, Tellus B Chem. Phys. Meteorol., 63, 107-116, 10.1111/j.1600-0889.2010.00513.x, 2011.

Drinovec, L., Gregorič, A., Zotter, P., Wolf, R., Bruns, E. A., Prévôt, A. S. H., Petit, J.-E., Favez, O., Sciare, J., Arnold, I. J., Chakrabarty, R. K., Moosmüller, H., Filep, A., and Močnik, G.: The filter-loading effect by ambient aerosols in filter absorption photometers depends on the coating of the sampled particles, Atmos. Meas. Tech., 10, 1043-1059, 10.5194/amt-10-1043-2017, 2017.

Fan, X., Cao, T., Yu, X., Wang, Y., Xiao, X., Li, F., Xie, Y., Ji, W., Song, J., and Peng, P.: The evolutionary behavior of chromophoric brown carbon during ozone aging of fine particles from biomass burning, Atmos. Chem. Phys., 20, 4593–4605, https://doi.org/10.5194/acp-20-4593-2020, 2020.

Gustafsson, O., Krusa, M., Zencak, Z., Sheesley, R. J., Granat, L., Engstrom, E., Praveen, P. S., Rao, P. S. P., Leck, C., and Rodhe, H.: Brown clouds over South Asia: Biomass or fossil fuel combustion?, Science, 323, 495-498, 10.1126/science.1164857, 2009.

Hammer, M. S., Martin, R. V., van Donkelaar, A., Buchard, V., Torres, O., Ridley, D. A., and Spurr, R. J. D.: Interpreting the ultraviolet aerosol index observed with the OMI satellite instrument to understand absorption by

organic aerosols: Implications for atmospheric oxidation and direct radiative effects, Atmos. Chem. Phys., 16, 2507-2523, 10.5194/acp-16-2507-2016, 2016.

Healy, R. M., Wang, J. M., Jeong, C. H., Lee, A. K. Y., Willis, M. D., Jaroudi, E., Zimmerman, N., Hilker, N., Murphy, M., Eckhardt, S., Stohl, A., Abbatt, J. P. D., Wenger, J. C., and Evans, G. J.: Light-absorbing properties of ambient black carbon and brown carbon from fossil fuel and biomass burning sources, J. Geophys. Res. Atmos., 120, 6619-6633, 10.1002/2015jd023382, 2015.

Hyvärinen, A.-P., Vakkari, V., Laakso, L., Hooda, R. K., Sharma, V. P., Panwar, T. S., Beukes, J. P., van Zyl, P. G., Josipovic, M., Garland, R. M., Andreae, M. O., Pöschl, U., and Petzold, A.: Correction for a measurement artifact of the Multi-Angle Absorption Photometer (MAAP) at high black carbon mass concentration levels, Atmos. Meas. Tech., 6, 81–90, https://doi.org/10.5194/amt-6-81-2013, 2013.

Jacobson, M. Z.: Strong radiative heating due to the mixing state of black carbon in atmospheric aerosols, Nature, 409, 695–697, https://doi.org/10.1038/35055518, 2001.

Jo, D. S., Park, R. J., Lee, S., Kim, S.-W., and Zhang, X.: A global simulation of brown carbon: Implications for photochemistry and direct radiative effect, Atmos. Chem. Phys., 16, 3413-3432, 10.5194/acp-16-3413-2016, 2016.

Kim, J., Bauer, H., Dobovičnik, T., Hitzenberger, R., Lottin, D., Ferry, D., and Petzold, A.: Assessing optical properties and refractive index of combustion aerosol particles through combined experimental and modeling studies, Aerosol Sci. Technol., 49, 340-350, 10.1080/02786826.2015.1020996, 2015.

Knox, A., Evans, G. J., Brook, J. R., Yao, X., Jeong, C. H., Godri, K. J., Sabaliauskas, K., and Slowik, J. G.: Mass absorption cross-section of ambient black carbon aerosol in relation to chemical age, Aerosol Sci. Technol., 43, 522-532, 10.1080/02786820902777207, 2009.

Mai, S., Ashwood, B., Marquetand, P., Crespo-Hernandez, C. E., and Gonzalez, L.: Solvatochromic effects on the absorption spectrum of 2-thiocytosine, J. Phys. Chem. B, 121, 5187-5196, 10.1021/acs.jpcb.7b02715, 2017.

Massabò, D., Caponi, L., Bernardoni, V., Bove, M. C., Brotto, P., Calzolai, G., Cassola, F., Chiari, M., Fedi, M. E., Fermo, P., Giannoni, M., Lucarelli, F., Nava, S., Piazzalunga, A., Valli, G., Vecchi, R., and Prati, P.: Multi-wavelength optical determination of black and brown carbon in atmospheric aerosols, Atmos. Environ., 108, 1-12, 10.1016/j.atmosenv.2015.02.058, 2015.

Meehl, G. A., Arblaster, J. M., and Collins, W. D.: Effects of black carbon aerosols on the Indian monsoon, J. Climate, 21, 2869– 2882, https://doi.org/10.1175/2007jcli1777.1, 2008.

Nakayama, T., Ikeda, Y., Sawada, Y., Setoguchi, Y., Ogawa, S., Kawana, K., Mochida, M., Ikemori, F., Matsumoto, K., and Matsumi, Y.: Properties of light-absorbing aerosols in the Nagoya urban area, Japan, in August 2011 and January 2012: Contributions of brown carbon and lensing effect, J. Geophys. Res. Atmos., 119, 12,721-712,739, 10.1002/2014jd021744, 2014.

Peng, J., Hu, M., Guo, S., Du, Z., Zheng, J., Shang, D., Zamora, M. L., Zeng, L., Shao, M., Wu, Y.-S., Zheng, J., Wang, Y., Glen, C. R., Collins, D. R., Molina, M. J., and Zhang, R.: Markedly enhanced absorption and direct radiative forcing of black carbon under polluted urban environments, Proc. Natl. Acad. Sci. USA, 113, 4266–4271, https://doi.org/10.1073/pnas.1602310113, 2016.

Phillips, S. M., and Smith, G. D.: Spectroscopic comparison of water- and methanol-soluble brown carbon particulate matter, Aerosol Sci. Technol., 51, 1113-1121, 10.1080/02786826.2017.1334109, 2017.

Qiu, Y., Wu, X., Zhang, Y., Xu, L., Hong, Y., Chen, J., Chen, X., and Deng, J.: Aerosol light absorption in a coastal city in Southeast China: Temporal variations and implications for brown carbon, J. Environ. Sci. (China), 80, 257-266, 10.1016/j.jes.2019.01.002, 2019.

Samset, B. H., Sand, M., Smith, C. J., Bauer, S. E., Forster, P. M., Fuglestvedt, J. S., Osprey, S., and Schleussner, C. F.: Climate impacts from a removal of anthropogenic aerosol emissions, Geophys. Res. Lett., 45, 1020-1029, 10.1002/2017gl076079, 2018a.

Schnaiter, M.: Absorption amplification of black carbon internally mixed with secondary organic aerosol, J. Geophys. Res. Atmos., 110, 10.1029/2005jd006046, 2005.

Sumlin, B. J., Pandey, A., Walker, M. J., Pattison, R. S., Williams, B. J., and Chakrabarty, R. K.: Atmospheric photooxidation diminishes light absorption by primary brown carbon aerosol from biomass burning, Environ. Sci. Technol. Lett., 4, 540-545, 10.1021/acs.estlett.7b00393, 2017.

Tóth, A., Hoffer, A., Nyirő-Kósa, I., Pósfai, M., and Gelencsér, A.: Atmospheric tar balls: Aged primary droplets from biomass burning?, Atmos. Chem. Phys., 14, 6669-6675, 10.5194/acp-14-6669-2014, 2014.

Updyke, K. M., Nguyen, T. B., and Nizkorodov, S. A.: Formation of brown carbon via reactions of ammonia with secondary organic aerosols from biogenic and anthropogenic precursors, Atmos. Environ., 63, 22-31, 10.1016/j.atmosenv.2012.09.012, 2012.

Wang, C.: Impact of anthropogenic absorbing aerosols on clouds and precipitation: A review of recent progresses, Atmos. Res., 122, 237-249, 10.1016/j.atmosres.2012.11.005, 2013.

Wang, J., Nie, W., Cheng, Y., Shen, Y., Chi, X., Wang, J., Huang, X., Xie, Y., Sun, P., Xu, Z., Qi, X., Su, H., and Ding, A.: Light absorption of brown carbon in Eastern China based on 3-year multi-wavelength aerosol optical property observations and an improved absorption Ångström exponent segregation method, Atmos. Chem. Phys., 18, 9061-9074, 10.5194/acp-18-9061-2018, 2018.

Wang, X., Heald, C. L., Sedlacek, A. J., de Sá, S. S., Martin, S. T., Alexander, M. L., Watson, T. B., Aiken, A. C., Springston, S. R., and Artaxo, P.: Deriving brown carbon from multiwavelength absorption measurements: Method and application to AERONET and Aethalometer observations, Atmos. Chem. Phys., 16, 12733-12752, 10.5194/acp-16-12733-2016, 2016.

Wei, X., Zhu, Y., Hu, J., Liu, C., Ge, X., Guo, S., Liu, D., Liao, H., and Wang, H.: Recent progress in impacts of mixing state on optical properties of black carbon aerosol, Curr. Pollut. Rep., 10.1007/s40726-020-00158-0, 2020.

Willis, M. D., Healy, R. M., Riemer, N., West, M., Wang, J. M., Jeong, C.-H., Wenger, J. C., Evans, G. J., Abbatt, J. P. D., and Lee, A. K. Y.: Quantification of black carbon mixing state from traffic: implications for aerosol optical properties, Atmos. Chem. Phys., 16, 4693–4706, https://doi.org/10.5194/acp-16-4693-2016, 2016.

Xie, C., Xu, W., Wang, J., Wang, Q., Liu, D., Tang, G., Chen, P., Du, W., Zhao, J., Zhang, Y., Zhou, W., Han, T., Bian, Q., Li, J., Fu, P., Wang, Z., Ge, X., Allan, J., Coe, H., and Sun, Y.: Vertical characterization of aerosol optical properties and brown carbon in winter in urban Beijing, China, Atmos. Chem. Phys., 19, 165-179, 10.5194/acp-19-165-2019, 2019.

Xie, X., Chen, Y., Nie, D., Liu, Y., Liu, Y., Lei, R., Zhao, X., Li, H., and Ge, X.: Light-absorbing and fluorescent properties of atmospheric brown carbon: A case study in Nanjing, China, Chemosphere, 251, 126350, 10.1016/j.chemosphere.2020.126350, 2020.

Yuan, Q., Xu, J., Wang, Y., Zhang, X., Pang, Y., Liu, L., Bi, L., Kang, S., and Li, W.: Mixing state and fractal dimension of soot particles at a remote site in the southeastern Ttibetan Plateau, Environ. Sci. Technol., 53, 8227-8234, 10.1021/acs.est.9b01917, 2019.

**References that were added,** as a result of the review process:

Andersson, A.: A model for the spectral dependence of aerosol sunlight absorption, ACS Earth Space Chem., 1, 533-539, 10.1021/acsearthspacechem.7b00066, 2017.

China, S., Mazzoleni, C., Gorkowski, K., Aiken, A. C., and Dubey, M. K.: Morphology and mixing state of individual freshly emitted wildfire carbonaceous particles, Nat. Commun., 4, 2122, 10.1038/ncomms3122, 2013.

McKay, G., Korak, J. A., Erickson, P. R., Latch, D. E., McNeill, K., and Rosario-Ortiz, F. L.: The case against charge transfer interactions in dissolved organic matter photophysics, Environ. Sci. Technol., 52, 406-414, 10.1021/acs.est.7b03589, 2018.

Trofimova, A., Hems, R. F., Liu, T., Abbatt, J. P. D., and Schnitzler, E. G.: Contribution of charge-transfer complexes to absorptivity of primary brown carbon aerosol, ACS Earth Space Chem., 3, 1393-1401, 10.1021/acsearthspacechem.9b00116, 2019.

**Responses to the comments of Referee #2**

We thank Referee #2 for the valuable comments, which have greatly helped us to improve the manuscript. Please find below our point-by-point responses (in blue normal font) to the comments of Referee #2 (*in black italic*). The changes in the revised manuscript are in green.

*This manuscript by Moschos et al. carefully described light absorption properties from various sites and sources. They also provide calculations of the optical properties. By combing the experimental and theoretical results, they provide useful conclusions regarding brown carbons and BC lensing effects. These results are useful for further studies regarding radiative forcing and climate. Thus, I recommend this manuscript to be published in ACP.*

We thank the referee for the positive feedback and useful comments, which we have addressed as explained in the responses given below.

*This study provides careful and comprehensive data and figures. All figures are carefully prepared and look nice. On the other hand, the figures include too much information, and I had a difficult time interpreting the meanings. For example, Fig 8 includes Y axis, X axis consisting of two parameters, color and shape of each plot, contour lines, and reference lines (dot line). Although I agree that the figures are useful and accurate, they are complicated to understand. It is just a suggestion, and the current figures are fine as is but may be improved by simplifying.*

Following the referee's comment, we have simplified whenever possible specific figures by removing excessive information. This includes a few references from Fig. 2, according to a comment from Referee #1, and unnecessary text appearing in the panels of Fig. 6 and Fig. 8, which has now been transferred to the respective captions. These revised figures are shown below. Also, Magadino in Fig. 8 (as well as in Fig. 4c and 7) is now shown with circles, to avoid potential confusion with the Zurich square markers.

[Figure]

Figure 2: Spectrally-resolved median *k* values (MAE in Fig. S10) of the different methanol-soluble (water-soluble: Moschos et al., 2018) offline AMS/PMF-based OA source components [(WINS-)BBOA corrected for extraction efficiency in MeOH; WINS-BBOA corrected for the water/methanol solvent effect; Other OA: HOA+COA+SOOA+fOOA], together with the IQR from different UV/Vis-PMF sensitivity runs (Text S4 & Table S2). The retrieved $k_{BrC}$ are compared to those of previous laboratory and field studies focusing on methanol-extractable or total organics [a: Chen & Bond (2010), b: Xie et al. (2017), c: Li et al. (2019), d: Liu et al. (2015), e: Lu et al. (2015), f: Adler et al. (2019), g: Corbin et al. (2019), h: Kirchstetter et al. (2004), i: Lack et al. (2012), j: Cappa et al. (2019), k: Yan et al. (2020), l: Cappa et al. (2012), m: Moschos et al. (2018)].

[Figure]

Figure 6: Optical closure results at six AE33 wavelengths (*x*-axis, linear scale), shown as cumulative contributions (labels) to total (calibrated) measured particulate absorption (*y*-axis) of the different carbonaceous aerosol species (bare BC, extractable BrC) and the BC absorption enhancement (lensing) defined as "remaining absorption". The total measured absorption, $b_{abs,total}$, is indicated with the orange circles (uncertainty: upward blue error bars). Note the different ranges of $b_{abs}$ values (*y*-axis) between the three cases. Orange arrows indicate the lensing contribution at longer wavelengths (660 nm, 880 nm) for the three cases, with $E_{abs,BC}$ being equal to (1.35 ± 0.10, 1.33 ± 0.10), (1.37 ± 0.24, 1.38 ± 0.17) and (1.35 ± 0.07, 1.33 ± 0.07) for Magadino winter, Magadino summer and Zurich, respectively. The error bars (minus direction shown; violet: BC; green: BC lensing; red: BrC) were computed by error propagation on the mean values [Eq. (C1)-(C3)] using the $C_\lambda$ (Fig. A1), $MAC_{bareBC,660nm}$ (Fig. S14), $AAE_{bareBC}$ (Fig. 4a) and $b_{abs,BrC-Mie}$ (Text S5) relative errors (Appendix C). The gray line indicates calculated total absorption by adding the absorption by BrC and absorption by BC including transparent shell lensing (assumed to be wavelength-independent) extrapolated from 660 nm.

[Figure]

Figure 8: Total measured aerosol AAE (referenced to 660 nm) vs. the ratio of calculated BrC absorption to measured total absorption at 370 nm (a) and 470 nm (b), showing the effect of BrC/BC interactions on the aerosol absorption profile. Uncertainty in the ratio $C_{370nm}/C_{660nm}$ used for the calibration of AE33 attenuation measurements translates into vertical error bars, while horizontal error bars include uncertainties in AE33 absolute calibration coefficients $C_{370nm}$ and in the BrC absorption at 370 nm from Mie calculations without contingency for errors associated with assuming homogeneous spheres. The isopleths indicate the extent of total absorption under- or over-estimation on the basis of a predicted absorption by BC and BrC, where (apparent) lensing by transparent BC coatings (using $AAE_{bare\ BC}$ = 0.93) is assumed to be wavelength-independent [isopleth = (measured – predicted) / predicted]. The dashed line, which corresponds to additive absorption if transparent shell lensing was increasing with decreasing wavelength (by 10 % or 6 % from 660 nm to 370 nm or 470 nm, respectively), serves to illustrate sensitivity of the closure to spectral extrapolation of transparent shell lensing. Over-predictions (negative isopleths) provide evidence for a lensing suppression effect.

*Specific comments:*

*Line 329: The reference by Alexander et al. (2008) is not adequate here because the literature does not show tar-balls but only discussed the possibility of tar-balls.*

The referee is correct. Considering also the discussion in the following sentences of the same paragraph including associated references, we have now removed the reference mentioned by the referee.

*Line 330: inorganic component. Fig S13 shows a Fe-bearing particle, which may not be a representative of "inorganic" particles.*

We thank the referee for this comment. Fe-bearing particles are primary, likely from abrasion processes (and dust), different from the secondary inorganic fraction (sulfate, nitrate, ammonium) that can influence the optical properties of BC and BrC through lensing. We have updated the caption of Fig. S13 with the following addition (the respective sentence of the main text was adjusted accordingly), and replaced the image and EDX spectrum with those of the adjacent particle.

"While the only spherical particles observed in untreated Zurich samples were non-carbonaceous, either Fe-bearing or containing K, Mg, Ca, Al and S (b), […]"

[Figure]

*Line 330: "Pseudospherical" can be a deformed particle that had been liquid in air and deformed on the filter when collected.*

We thank the referee for this suggestion, which we have added in the caption of the FE-SEM/EDX figure.

**Responses to the comments of Referee #3**

We thank anonymous Referee #3 for this critical review. Please find below our point-by-point responses (in blue normal font) to the comments of Referee #3 (*in black italic*).

*The manuscript attempts to resolve almost all issues of carbonaceous light absorption based on annual filter-based measurements at two locations. It relies an immense number of references just as a review paper, but in this case the references are actively engaged in the discussion of the methodology applied. That makes the manuscript extremely difficult to follow and understand. Despite its complexity, the manuscript has little if any novelty, it is like a demonstration of all techniques related to filter-based absorption measurements over the past 20 years.*

We disagree with the referee regarding the novelty of our study. This is not a demonstration of all techniques related to filter-based absorption measurements over the past twenty years. Our study combines multiple measurements to answer a specific scientific question: how do BrC, BC and lensing-inducing coatings together affect the aerosol absorption and absorption profile? We consider all measurements and results to be necessary to arrive to the study conclusions, in Figures 8 and 9.

Analyzed aerosol samples are from typical Alpine and urban background locations covering at least a full year, providing access to the most important sources of primary and aged aerosols and covering a wide range of aerosol properties. With this, we are able to quantify the importance of the different aerosol components for the total absorption under different atmospheric conditions, and to assess how their complex interactions affect the aerosol absorption profile. We consider our results to be of general interest for atmospheric chemistry and physics, including optical modelling, the source apportionment community and global modelling, as mentioned in the revised abstract (see above).

We believe that the paper follows a logical thread and is balanced between the different inter-linked topics addressed. Despite the existing links between the results, each subsection has a title that helps the reader to assess whether there is any take-home message in the content to spend time reading it. The referee criticized the great number of citations we have used. As detailed above in the response to Referee #1, we have removed 27 % of the citations appearing in the preprint.

*Instead of using the more accepted concept of light absorbing carbon continuum, the manuscript relies on the simplified concept of BrC vs BC with spectrally resolvable absorption properties. This is a simple yet quite an established methodology for estimating BB vs FF contributions to PM with all its inherent biases and uncertainties. Due to the latter, this filter-based approach can by no means yield results that may be used for proving a hypothesis regarding aerosol mixing state and morphology. The statement that the 'first experimental evidence is provided for the suppression of lensing effect by BrC' is simply not supported by the optical closure calculations at shorter wavelength between solvent-based and filter-based optical measurements. Both techniques have a number of limitations and uncertainties exhaustively discussed in the scientific literature, and both are based on several simplifying assumptions which may not necessarily be valid. In the light of these facts, the residual term of optical closure calculations between two fundamentally different measurement methods can by no means signify any 'lensing effect'. The authors themselves devote detailed discussions to (even non-quantifiable) uncertainties and simplifying assumptions (some of them can equally be biases), yet they do not come to the conclusion that a small residual term in closure calculations is well within the range of uncertainties and should not be over-interpreted.*

We disagree with the reviewer that the light-absorbing carbon continuum concept has gained more acceptance than the BrC/BC model, but that was not the main point of this comment. The main point of this comment was that the lensing suppression arguments are somewhat speculative.

We have acknowledged that lensing suppression is not fully constrained by our calculations and sought to avoid over-interpretation. This is the reason why we have devoted a detailed discussion on unquantifiable uncertainties and clearly stated that the observed effect needs to be confirmed through dedicated laboratory studies. We observe that at 370 nm the measured absorption is systematically (Fig. 6) and beyond the quantifiable uncertainties (Fig. 8) lower than the predicted absorption of

individual components. We have used widespread techniques to arrive at these conclusions, so other studies using such techniques may also see similar effects. Therefore, we wanted to comment on the lensing suppression point, which is one of multiple points made in the manuscript. We believe that the new abstract (see above) reflects well our view point.

*Optical closure calculations are based on the assumption that total filter absorption, absorptions in methanol and water extract, and residual absorption on filter (after solvent extraction) are all additive. Is it possible that solvent extraction changes some properties on filter that affect absorption measurements (e.g. scattering effect)? Is there any hysteresis of solvent extraction (e.g. residual water that affects measurements)? Additivity means that if a spot of a loaded filter is measured for absorption, then extracted in water and methanol, then the extracts are uniformly redispersed on the residual filter spot and the solvents are evaporated, and the filter spot is again measured for absorption, we get exactly the same absorption as for the untreated filter. I never did such a simple experiment but I would seriously doubt that the two values would be identical.*

The assumptions listed (additive absorption, solvent effects, hysteresis effects) have been addressed by previous work, from our group (e.g. Corbin et al., 2019) and from others. While the solvent treatment may affect the BC morphology, changes in absorption upon morphological changes of BC have been shown to be negligible in the Rayleigh–Debye–Gans approximation (Radney et al., 2014). In addition, previous experimental work has demonstrated consistency between solvent (methanol) extraction and Aethalometer measurements (Liu et al., 2013). Other studies have applied a similar "BC de-coating" approach to infer both the optical properties and lensing effect for BC from filter-based measurements, e.g., OC/EC analyzer (e.g., Cui et al., 2016; Chen et al., 2017). However, we agree with the reviewer that caution is warranted around these assumptions. This is why in our analysis we have only considered these measurements to determine solely the absorption wavelength-dependence (AAE) of the residual (insoluble) material. This is more accurate than residual absolute absorption values, which are not as reproducible due to mechanical removal of BC particles (as stated in the SI). The absolute absorption or MAC of bare BC are inferred from the intercept of Fig. 4. Residual AAE values obtained using the extraction approach are consistent with previous experimental and modelling values for bare BC (e.g., Lack and Cappa, 2010).

**References** (not included in the preprint)

Chen, B., Bai, Z., Cui, X., Chen, J., Andersson, A., and Gustafsson, O.: Light absorption enhancement of black carbon from urban haze in Northern China winter, Environ. Pollut., 221, 418-426, 10.1016/j.envpol.2016.12.004, 2017.

Cui, X., Wang, X., Yang, L., Chen, B., Chen, J., Andersson, A., and Gustafsson, O.: Radiative absorption enhancement from coatings on black carbon aerosols, Sci. Total Environ., 551-552, 51-56, 10.1016/j.scitotenv.2016.02.026, 2016.

Radney, J. G., You, R., Ma, X., Conny, J. M., Zachariah, M. R., Hodges, J. T., and Zangmeister, C. D.: Dependence of soot optical properties on particle morphology: Measurements and model comparisons, Environ. Sci. Technol., 48, 3169-3176, 10.1021/es4041804, 2014.

---

## Author Response (AR2)

**Responses to the comments of the Editor**

We thank the Editor for the valuable comments, which have greatly helped us to further improve the manuscript. Please find below our point-by-point responses (in blue normal font) to your feedbacks (*in black italic*). The changes in the revised manuscript are in green.

*Dear Authors,*

*I have reviewed the comments from three referees and your responses. While two of them recommended this work to be published in Atmospheric Chemistry and Physics, Referee #3 has pointed out the technical issues for quantifying BC absorption enhancement due to lensing effect using the filter-based measurement techniques that can affect the subsequent argument regarding suppression of lensing effect by BrC at the shorter wavelength. Overall, I find this manuscript presents valuable observations to advance our understanding on atmospheric BC and BrC absorptions and to address the potential impacts of BC-BrC interactions on total aerosol absorptions. Nevertheless, I would like to make two major comments on the revised version that are related to some of the comments from Referee #3 before considering this work to be published in Atmospheric Chemistry and Physics.*

We thank the Editor for the evaluation of our revised manuscript and constructive comments on remaining issues. We are confident that we have fully addressed these issues and hope that the modified version of the manuscript can be accepted in ACP.

*1) Section 3.5: The authors speculate that the observed BC absorption enhancement (Eabs) is primarily due to the lensing effect of BC coatings (line 365). However, it is possible that only a fraction of BC coating proxy (e.g., OOA+BBOA+NO3+SO4) are internally mixed with airborne BC. I suggest the authors to further comment whether the filter-based Eabs can be also caused by those chemical components that are externally mixed with BC and perhaps other factors such as calibration uncertainties and particle morphology. If so, the filter-based BC lensing effects (or Eabs) can be fundamentally different from the "true" lensing effects for airborne BC, and hence the two terms should not be interchangeable although the filter-based Eabs values are not significantly different from those reported in the literature based on in-situ measurements (lines 395-398). If this is the case, it is strongly recommended to use the operational-defined term "Filter-based Eabs" all the time (including the sub-heading) to avoid over-interpretation of data/misleading conclusion.*

We appreciate the Editor's comment. Indeed, only a fraction of the proxy is expected to be internally mixed with BC, as not all particles contain a BC "core". This is already stated in lines 396-397 (in the newly revised manuscript), but does not affect our conclusions.

The Editor questions whether $E_{abs}$ can be affected by uncertainties in the calibration of our filter-based measurements or be caused by the same non-refractory chemical components, but that are externally mixed with BC. We have now investigated the dependence of the calibration coefficients, $C_{660nm}$, on the proxy, for 18 $PM_{10}$ samples measured with MWAA, and found no relationship: the Pearson's $r$ is -0.08 and the slope is not significantly different from zero ($p = 0.74$). This suggests that uncertainties in the calibration might not affect the relationship between filter-based $E_{abs}$ and the proxy. We cannot fully exclude the contribution of externally mixed non-refractory species deposited on the filters to the $E_{abs}$, even if the $E_{abs}$ values we found are consistent with those determined based on in-situ measurements (lines 405-406 in the newly revised manuscript). Therefore, we have followed the Editor's suggestion and added "filter-based" next to lensing or $E_{abs}$ throughout the manuscript text (including sub-headings), in order to provide a clear distinction between in-situ- and filter-based lensing and to avoid over-interpretation. The second paragraph of Sect. 3.5. has been modified as follows:

Figure 4c presents $MAC_{BC}$ as a function of a proxy for the BC coating thickness, i.e., the ratio between the combined concentrations of major-SIA, OOA and BBOA and EC (NR-PM/EC; Table S4). While the variability in $MAC_{BC}$ is not driven by the EC sources (Fig. 4b), $MAC_{BC}$ increases linearly with NR-PM/EC < 33 consistently, unlike for other tested proxies (including the total OA mass, OA:EC, OOA:OA, OOA:EC or Sulfate:EC), indicating a filter-based BC lensing effect due to coating by multiple non-refractory components. The presence of coated BC particles is supported by the observation of

compacted BC particles from SEM measurements (Fig. S13). We have examined the relationship between $C_{660nm}$ and the proxy and found them to be independent, indicating that uncertainties in the Aethalometer calibration do not affect the resulting relationships between $E_{abs}$ and the proxy. While we attribute the filter-based (apparent) BC absorption enhancement to lensing, future studies should evaluate its potential dependence on chemical components that are externally mixed with BC, including tar-balls absorbing at longer wavelengths (Sect. 3.4), as well as calibration uncertainties and/or the deposited particle morphology.

*2) Section 3.6 and 3.7: While the suppression of filter-based BC lensing induced by BrC coatings is a novel observation that can provide important insight into our understanding of BC-BrC interaction and total aerosol absorption as well as uncertainties of filter-based measurements for quantifying BrC absorption, I agree with Referee #3 that the closure calculation involves many assumptions and simplified concepts that may lead to over-interpretation of results. Overall, I think it is still important to report the possibility of filter-based BC lensing suppression due to BrC coatings and such data analysis approach to the scientific communities for future research but the authors should tone down some of the related arguments/conclusion throughout the manuscript. Both quantifiable and non-quantifiable uncertainties described in Appendix C should be clearly presented in the main text to ensure the reader can easily recognize the major uncertainties and limitation of the closure calculation.*

*Alex*

We agree with the Editor. as we had done in the revised abstract, we have now replaced the word "evidence" by "indication" throughout the main text when discussing the "inferred" filter-based lensing suppression at shorter wavelengths. While quantifiable uncertainties are already discussed in Sect. 3.7 together with Fig. 8 (lines 474-476 in the newly revised manuscript), we have now added in the end of Sect. 3.7 a sentence commenting also on potential unquantifiable uncertainties and referring to Appendix C for a more detailed discussion:

"[…] Finally, we note that our optical closure is limited in terms of interpretation of lensing effects, due to unquantifiable uncertainties potentially associated with filter sampling artifacts, possible chemical interactions between airborne BrC molecules or with BC, and the use of simplified Mie calculations to obtain the particulate BrC absorption (Appendix C)."

We have also added in Sect. 4 the following sentence, to provide a more balanced discussion on the implications of our findings as suggested by the Editor:

"While the optical absorption closure approach presented here involves multiple assumptions and simplified concepts, our results provide useful experimental insights into understanding BrC/BC interactions and total atmospheric aerosol absorption, as well as uncertainties of filter-based measurements for quantifying BrC absorption. If lensing suppression occurs due to internal mixing of BC and BrC as is apparently the case for many samples in our study […]"